# PTEN regulates starburst amacrine cell dendrite morphology during development

Teva W. Bracha[1], Nina Luong[1], Joseph Leffler[2], Benjamin Sivyer[2,*] and Kevin M. Wright[3,‡]

## ABSTRACT

Neurons are subject to extensive developmental regulation to ensure precise subtype-specific morphologies that are intimately tied to their function. Starburst amacrine cells (SACs) in the mammalian retina have a highly stereotyped, radially symmetric dendritic arbor that is essential for their role in direction-selective circuits in the retina. We show that PTEN, the primary negative regulator of the PI3K-AKT-mTOR pathway that is highly implicated in neurodevelopmental disorders, regulates SAC morphology in a cell-autonomous manner. *Pten*-deficient mouse SACs show a nearly twofold increase in the number of dendritic branches, while other morphological properties remain largely unchanged. These morphological changes arise late in SAC development, after dendrite development is largely complete, and persist into adulthood. Mechanistically, excessive dendritic branching appears to arise from dysregulated mTOR activity. Despite this increase in dendritic branches, *Pten*-deficient SACs maintain a normal population number, normal organization of synaptic outputs and intact direction-selectivity in the retina. Collectively, these results show that PTEN is essential in mouse for the normal development of highly stereotyped neuronal morphology.

KEY WORDS: Neuron, Retina, Dendrite, PTEN, Morphology

## INTRODUCTION

Since Ramón y Cajal, neuroscientists have appreciated the complexity of the nervous system and the vast array of neuronal shapes and sizes. This morphological diversity underlies the computational power of the nervous system, as neurons acquire specific morphologies that are uniquely adapted to support their function (Lefebvre et al., 2015). Morphological features can be used in conjunction with molecular profiles and functional properties to classify neurons into discrete subtypes (Zeng and Sanes, 2017). Neuronal morphology is influenced by intrinsic factors and extrinsic cues in the extracellular environment. Cell type-specific transcription factors control the expression of effectors that, in turn, regulate the morphological development of a neuronal subtype

(Jan and Jan, 2010). A neuron's complement of cell surface receptors allows it to organize dendritic arbors and synaptic partners in response to extrinsic cues. These receptors converge on intracellular signaling cascades that modulate cytoskeletal dynamics, leading to differences in neurite elongation, branch initiation and stabilization (Lefebvre, 2021).

Many insights about the development of neuronal morphology come from stereotyped neuronal subtypes in a range of model organisms. PVD neurons in *C. elegans* have a defined pitchfork-like projection pattern that line the body wall and are fundamental to mechanosensation and proprioception (Albeg et al., 2011). Forward genetic screens have identified transcription factors, cell surface receptors and intracellular signaling molecules that are required for the stereotyped PVD neuron morphology (Oren-Suissa et al., 2010; Tsalik et al., 2003; Inberg et al., 2019; Sundararajan et al., 2019). The dendritic arborization (da) neurons in *Drosophila* larvae can be easily distinguished into four distinct subtypes based on the degree of dendrite branching (Grueber et al., 2002). The morphological complexity of the four da neuron subtypes is determined by relative levels of three transcription factors: abrupt, cut and knot (Jinushi-Nakao et al., 2007; Sugimura et al., 2004; Grueber et al., 2003). Purkinje neurons in the cerebellar cortex have large, planar dendritic arbors with extensive branching that maintain a high degree of self-avoidance. Multiple pathways govern the development of these arbors, including repulsive Slit/Robo signaling, protocadherin-mediated self-avoidance, and actin regulators Daam1 and MTSS (Gibson et al., 2014; Lefebvre et al., 2012; Kawabata Galbraith et al., 2018).

Starburst amacrine cells (SACs) in the mammalian retina are an excellent model for studying the development of neuronal morphology due to their stereotyped radially symmetric branching pattern, defined circuit function, and the link between their dendritic form and neuronal function (Morrie and Feller, 2016; Prigge and Kay, 2018). SAC somas reside in the inner nuclear layer (INL) and ganglion cell layer (GCL), and project their dendrites to the inner plexiform layer (IPL), where they form planar dendritic arbors that stratify in sublaminas 2 (S2) and 4 (S4), respectively (Fig. 1A). Work from multiple labs has identified cell surface receptors that direct SAC morphology and stratification in the IPL. MEGF10 regulates SAC mosaic spacing through mediation of homotypic contacts during development (Kay et al., 2012; Ray et al., 2018; Kozlowski et al., 2024). Repulsive signaling mediated by FLRT2/UNC5 regulates SAC dendrite stratification (Prigge et al., 2023). Bi-directional plexin A2 and semaphorin 6A signaling is crucial for SAC radial morphology and dendrite stratification (Sun et al., 2013; James et al., 2024). Other studies have identified molecules that disrupt SAC dendrite morphology without affecting stratification, suggesting that these are separate processes. γ-Protocadherins (γ-Pcdhs) undergo alternative splicing to generate hundreds of isoforms, with homophilic matching between isoforms mediating self-recognition in SAC dendrites. Genetic deletion of all γ-Pcdh isoforms causes SAC dendrites to fasciculate, disrupting their radial morphology; expression

[1]Neuroscience Graduate Program, Oregon Health & Science University, Portland, OR 97239, USA. [2]Casey Eye Institute, Oregon Health & Science University, Portland, OR 97239, USA. [3]Vollum Institute, Oregon Health & Science University, Portland, OR 97239, USA.
*Present address: University of Sydney, Darlington, NSW 2006, Australia.

‡Author for correspondence (wrightke@ohsu.edu)

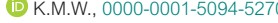 K.M.W., 0000-0001-5094-5270

of a single γ-Pcdh isoform in γ-Pcdh-deficient SACs is sufficient to restore self-recognition and normal radial morphology (Lefebvre et al., 2012). Loss of the cell-surface protein AMIGO2 results in a 1.5-fold increase in the size of SAC dendritic arbors, but does not affect their stereotyped branching, symmetry or stratification (Soto et al., 2019). While these cell surface proteins are crucial for regulating SAC morphology, little is known about the downstream intracellular signaling pathways that govern SAC morphology.

PTEN (phosphatase and tensin homologue) is a protein and lipid phosphatase that canonically functions as the primary negative regulator of the PI3K-AKT-mTOR pathway (Worby and Dixon, 2014). This pathway functions downstream of cell surface receptors to regulate neuronal differentiation, migration, neurite outgrowth and survival (Rademacher and Eickholt, 2019). PTEN regulates neurite growth and branching in mammalian neurons *in vivo*. Deletion of *Pten* results in neuronal hypertrophy and increased dendrite branching in cortical pyramidal neurons, dentate granule cells, serotonergic raphe neurons and Purkinje neurons (Chen et al., 2021; Gallent and Steward, 2018; Cupolillo et al., 2016; Kwon et al., 2006). This can ultimately lead to altered synaptic connectivity and neuronal hyperexcitability (Luikart et al., 2011; Santos et al., 2017).

Whether PTEN plays a role in regulating the highly stereotyped dendritic morphology of SACs remains an open question. Deletion of *Pten* from retinal progenitors results in widespread defects in neuronal differentiation, migration, cellular lamination, mosaic spacing and dendrite stratification throughout the retina, precluding analysis of SAC morphology (Sakagami et al., 2012; Cantrup et al., 2012; Tachibana et al., 2016; Touahri et al., 2024). We therefore used a $ChAT^{Cre}$ line to delete *Pten* specifically from post-migratory SACs ($ChAT^{Cre};Pten^{cKO}$) to address its cell-autonomous role in regulating SAC morphology. SACs in $ChAT^{Cre};Pten^{cKO}$ mice had a >1.5-fold increase in dendritic branching without affecting the overall length or field area of their dendritic arbors. We found that these branching phenotypes arise gradually during the later phase of SAC dendrite development and persist into adulthood. Analysis of signaling pathways downstream of PI3K-AKT suggests that increased dendrite branching is likely due to increased mTOR activity. Finally, we show that loss of *Pten* does not affect the compartmentalization of synaptic outputs in SACs or the function of the direction-selective circuit in the retina.

## RESULTS

### SAC-specific deletion of *Pten* does not affect cell density, somal lamination, mosaic spacing or dendrite stratification

Pan-retinal deletion of *Pten* from retinal progenitors causes abnormal somal lamination, mosaic spacing and dendrite stratification in retinal neurons, including SACs (Cantrup et al., 2012; Sakagami et al., 2012). Subsequent work identified a role for PTEN in regulating the vesicular trafficking of cell-adhesion molecules that are involved in establishing retinal neuron mosaics and dendrite stratification (Touahri et al., 2024). However, it is unclear whether these defects reflect a SAC-autonomous effect or are due to the overall disorganization of the retina. To circumvent this confounder, we used a $ChAT^{Cre}$ line to selectively drive recombination in SACs beginning at postnatal day 1 (P1) (Ray et al., 2018). This timing coincides with the end of SAC laminar migration and the initiation of their dendritic stratification in the nascent IPL.

To better understand which aspects of SAC development require cell-autonomous PTEN function, we conducted a side-by-side comparison of pan-retinal ($Six3^{Cre}$) and SAC-specific ($ChAT^{Cre}$) *Pten* conditional knockouts. Using retinal flat mount preparations

from P28 $Six3^{Cre};Pten^{cHet}$ and $Six3^{Cre};Pten^{cKO}$ mice, we confirmed that, while all cells in the GCL were positive for PTEN in $Six3^{Cre};Pten^{cHet}$ retinas, staining was completely absent in $Six3^{Cre};Pten^{cKO}$ retinas (Fig. 1B-C′). In contrast, PTEN was selectively lost from SACs in $ChAT^{Cre};Pten^{cKO}$ retinas (Fig. 1D-E′), but retained in all other GCL neurons (Fig. 1D-E′). We confirmed PTEN loss from SACs by measuring soma size, as neuronal hypertrophy is consistently seen after *Pten* deletion. In both $Six3^{Cre};Pten^{cKO}$ and $ChAT^{Cre};Pten^{cKO}$ retinas, SAC soma sizes were significantly increased compared to their respective controls (Fig. 1F-I).

We next compared the early developmental processes of differentiation, migration and mosaic spacing in $Six3^{Cre};Pten^{cKO}$ and $ChAT^{Cre};Pten^{cKO};Ai9$ retinas. Since loss of a single *Pten* allele can affect neuronal differentiation in certain contexts, we included both wild-type and heterozygous controls (Clipperton-Allen and Page, 2014; Chen et al., 2015; Fernandez et al., 2025 preprint). Consistent with previous studies using pan-retinal deletion of *Pten*, we found reduced cellular density and mosaic regularity of SACs in $Six3^{Cre};Pten^{cKO}$ retinas compared to controls (Fig. 2A-E) (Cantrup et al., 2012; Sakagami et al., 2012). In contrast, there was no difference in cellular density or mosaic spacing of SACs following $ChAT^{Cre}$-mediated *Pten* deletion (Fig. 2F-J). The lack of differentiation or migration phenotypes in $ChAT^{Cre};Pten^{cKO};Ai9$ retinas is likely due to *Pten* deletion occurring after these developmental processes are nearly complete and allows us to examine its cell-intrinsic role during dendrite development without these confounders.

To examine how deletion of *Pten* affects SAC dendrite stratification in the IPL, we stained retinal cross-sections at P28. Like previous studies that examined pan-retinal deletion of *Pten*, $Six3^{Cre};Pten^{cKO}$ mice had disrupted retinal architecture and SAC lamination, with highly disorganized S2 and S4 bands (Fig. 2K-N, Fig. S1A-C). In contrast, we observed two well-defined tdTomato$^+$ bands corresponding to S2 and S4 in $ChAT^{Cre};Pten^{cKO};Ai9$ retinas (Fig. 2O-R, Fig. S1D-F). While these bands appeared slightly less compact in $ChAT^{Cre};Pten^{cKO};Ai9$ retinas compared to littermate controls, there was no statistical difference between genotypes. Therefore, PTEN is not required for SAC dendrite stratification in the IPL.

### Loss of *Pten* causes increased dendritic branching in SACs
SACs have a high degree of dendritic overlap with their neighbors, preventing analysis of individual cell dendrites at a population level. We therefore sparsely labeled SACs by injecting a *Cre*-dependent AAV (AAV8-FLEx-tdTomato-CAAX) into the vitreous at P1-P2. We analyzed $ChAT^{Cre};Pten^{WT}$, $ChAT^{Cre};Pten^{cHet}$ and $ChAT^{Cre};Pten^{cKO}$ SACs from both the GCL and INL at P21, when dendrite morphology is largely mature (Fig. 3A-C″) (Sun et al., 2013). We quantified total dendritic length, number of branch points, dendritic field area and dendritic self-crossings (Fig. 3D-G). While we did not detect any differences between $ChAT^{Cre};Pten^{WT}$ and $ChAT^{Cre};Pten^{cHet}$ SACs, there were significant changes in $ChAT^{Cre};Pten^{cKO}$ SACs. There was a small increase in total dendritic length in GCL SACs but not in INL SACs in $ChAT^{Cre};Pten^{cKO}$ retinas (Fig. 3D). The total number of branch points was significantly increased in both INL and GCL SACs in $ChAT^{Cre};Pten^{cKO}$ retinas, nearly doubling in number (Fig. 3E). Despite the increase in dendrite branching, $ChAT^{Cre};Pten^{cKO}$ SACs show no changes in their dendritic field size (Fig. 3F). We also found that $ChAT^{Cre};Pten^{cKO}$ SACs have a significant increase in dendritic self-crossings compared to controls (Fig. 3A″-C″,G). Using a Sholl analysis to measure local changes in dendritic density, we found that $ChAT^{Cre};Pten^{cKO}$ GCL SACs showed relatively localized increases in dendritic density in the distal 50% of their dendritic arbor (Fig. 3H), whereas

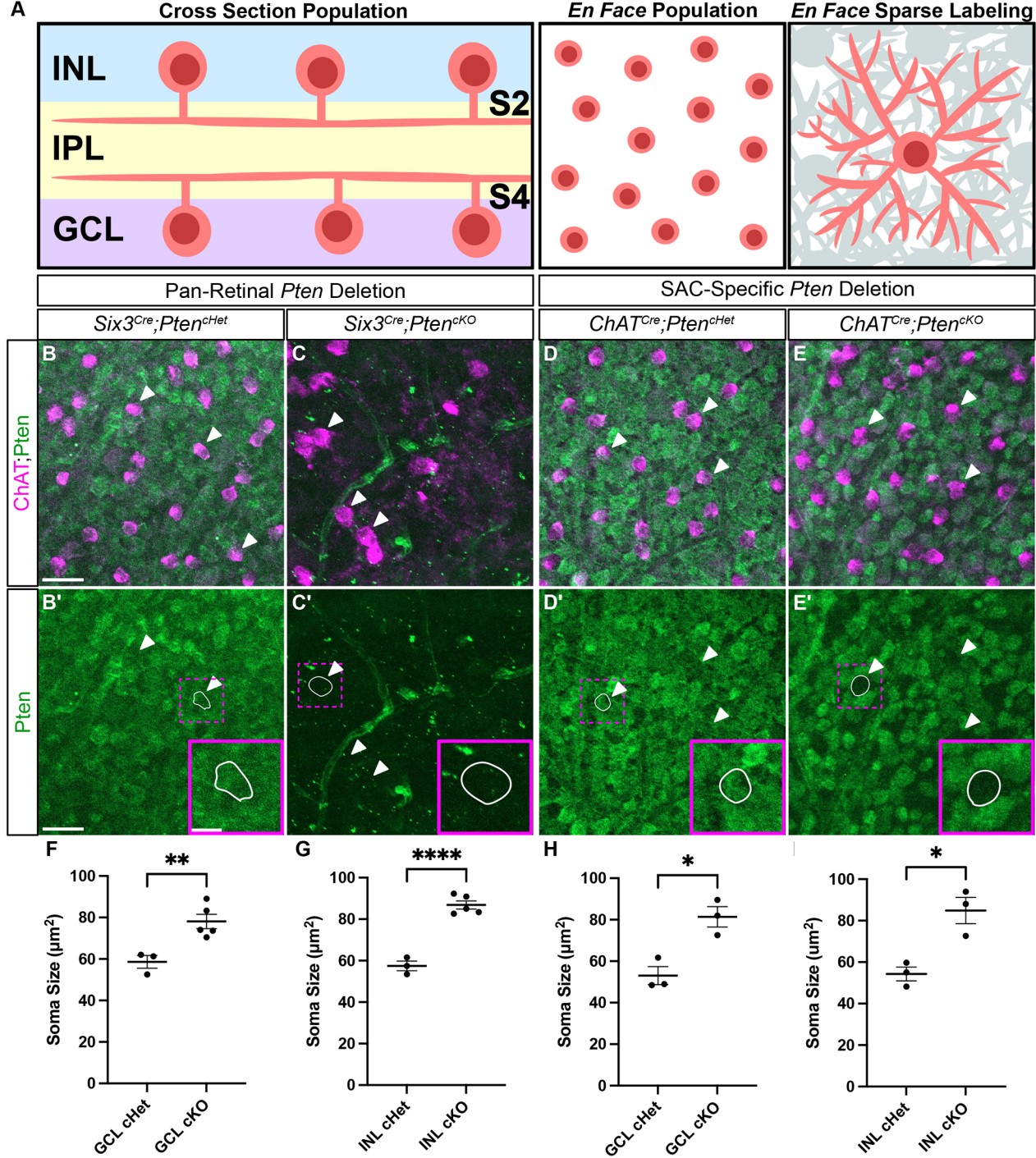

**Fig. 1. Validation of SAC specific *Pten* deletion.** (A) Schematic showing different retinal preparations for visualizing SACs. Retinal cross-sections (left panel) are used to analyze cellular lamination and dendrite stratification. Retinal flat mounts imaged in an *en face* preparation are used for population measurements (middle panel) and single cell morphology (right panel). (B-E′) P28 retinal flat mounts immunostained for ChAT (magenta) to label SAC somas and for PTEN (green) show that PTEN is present in all cells in the ganglion cell layer (GCL). In *Six3^{Cre};Pten^{cKO}* retinas, PTEN is eliminated from all GCL cells (C,C′), whereas in *ChAT^{Cre};Pten^{cKO}* retinas, PTEN is selectively lost only from SACs (white arrowheads) (E,E′). The individual SACs shown in the insets outlined in white. (F-I) Quantification of SAC soma sizes at P28 reveals somal hypertrophy, a common phenotype seen after *Pten* deletion, in both *Six3^{Cre};Pten^{cKO}* and *ChAT^{Cre};Pten^{cKO}* GCL and inner nuclear layer (INL) SACs (cHet: $n=3$, 58.62±3.05; cKO: $n=5$, 78.08±3.478, $P=0.0433$ in F; cHet: $n=3$, 57.48±2.346; cKO: $n=5$, 86.83±1.985, $P=0.0017$ in G; cHet: $n=3$, 53.05±4.324; cKO: $n=3$, 81.35±4.912, $P=0.0124$ in H; cHet: $n=3$, 54.32±3.331; cKO: $n=3$, 84.87±6.359, $P=0.0307$ in I). Data are mean±s.e.m. Two-tailed unpaired Student's *t*-test. Scale bars: 25 µm; 10 µm (insets).

*ChAT^{Cre};Pten^{cKO}* INL SACs showed a generalized increase in dendritic density across their entire arbor (Fig. 3I). Analysis of branching level showed that *ChAT^{Cre};Pten^{cKO}* SACs in the GCL developed more lower level branches, while *ChAT^{Cre};Pten^{cKO}* SACs in the INL had an increase in higher level branches (Fig. S2A,B). These results show that, while PTEN is not required for establishing arbor size in SACs, it regulates proper dendrite branching. These local changes in density are significant as SACs are purely dendritic neurons with spatially

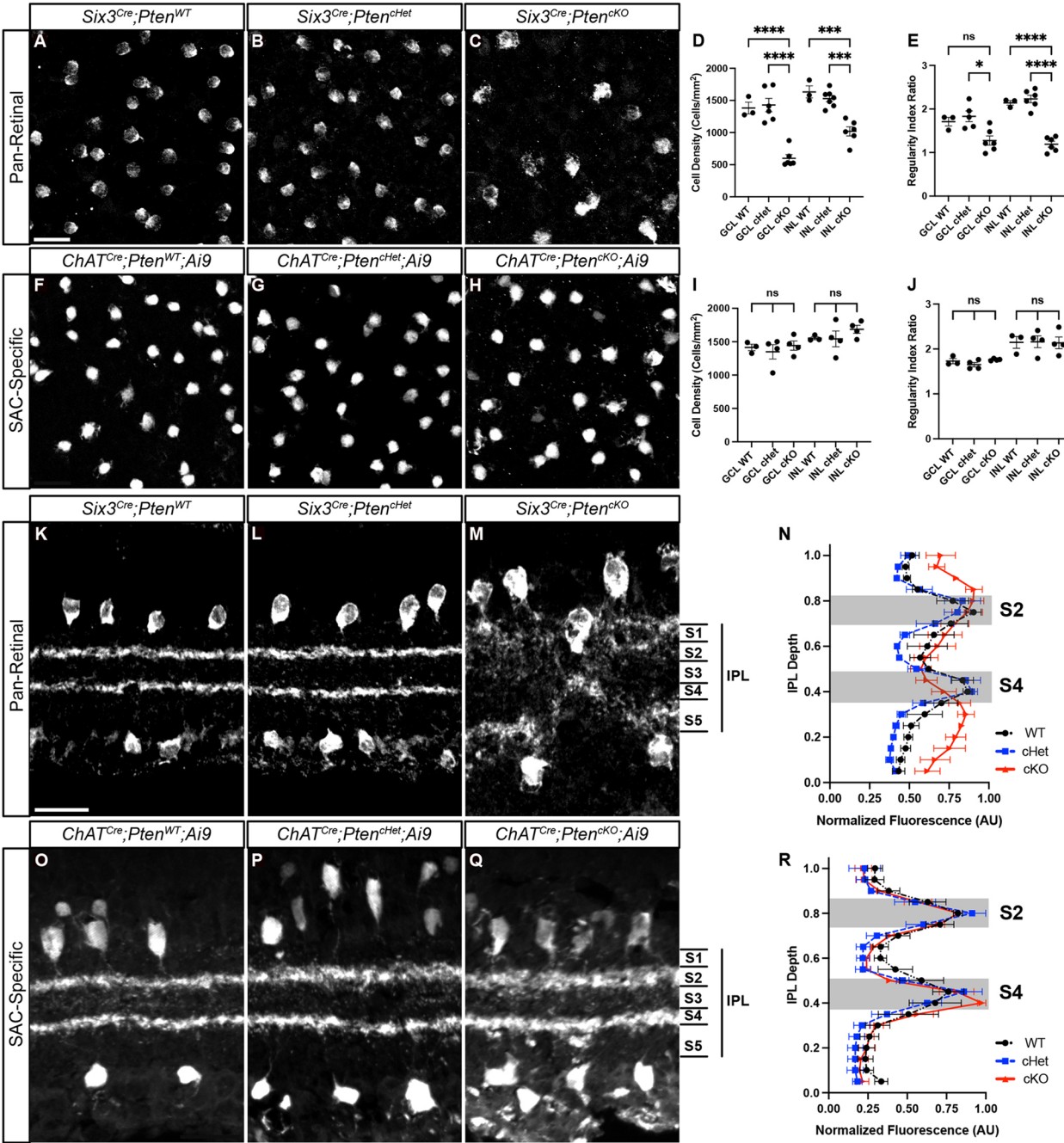

**Fig. 2. Selective deletion of *Pten* from SACs does not affect their cell density, mosaic spacing or dendrite lamination.** (A-C) Images of P28 *Six3^Cre; Pten^WT*, *Six3^Cre;Pten^cHet* and *Six3^Cre;Pten^cKO* retina flat mounts with ganglion cell layer (GCL) SACs labeled by ChAT immunostaining. (D,E) Quantification of cell density (D; GCL WT: *n*=3, 1.71±0.097; GCL cHet: *n*=5, 1.83±0.12; GCL cKO: *n*=6, 1.27 ±0.11; INL WT: *n*=3, 2.14±0.053; INL cHet: *n*=6, 2.23±0.085; INL cKO: *n*=6, 1.19±0.068) and mosaic spacing (E; GCL WT: *n*=3, 1382±90.79; GCL cHet: *n*=6, 1428±103.5; GCL cKO: *n*=6, 598.6±58.35; INL WT: *n*=3, 1633 ±94.77; INL cHet: *n*=7, 1530±48.96; INL cKO: *n*=6, 1019±71.13) of GCL and inner nuclear layer (INL) SACs shows decreased cell density and mosaic regularity in *Six3^Cre;Pten^cKO* retinas following pan-retinal deletion of *Pten* [*P*<0.0001 for GCL and INL (D), *P*=0.01 for GCL and *P*<0.0001 for INL (E); one-way ANOVA with post-hoc Tukey HSD test]. (F-H) Images of P28 *ChAT^Cre;Pten^WT;Ai9*, *ChAT^Cre;Pten^cHet;Ai9* and *ChAT^Cre;Pten^cKO;Ai9* retina flat mounts with GCL SACs labeled by tdTomato. (I,J) Quantification shows normal cell density (I; GCL WT: *n*=3, 1415±47.91; GCL cHet: *n*=4, 1349±107.4; GCL cKO: *n*=4, 1442 ±69.42; INL WT: *n*=3, 1553±23.73; INL cHet: *n*=4, 1542±117.8; INL cKO: *n*=4, 1686±59.67) and mosaic spacing (J; GCL WT: *n*=3, 1.73±0.059; GCL cHet: *n*=4, 1.65±0.048; GCL cKO: *n*=4, 1.76±0.011; INL WT: *n*=3, 2.15±0.13; INL cHet: *n*=4, 2.16±0.13; INL cKO: *n*=4, 2.14±0.13) of SACs in *ChAT^Cre;Pten^cKO;Ai9* retinas following selective deletion of *Pten* from SACs [*P*=0.7195 for GCL and 0.4336 for INL (I), *P*=0.1871 for GCL and 0.9901 for INL (J); one-way ANOVA with post-hoc Tukey HSD]. (K-M) P28 *Six3^Cre;Pten^WT*, *Six3^Cre;Pten^cHet* and *Six3^Cre;Pten^cKO* retina cross-sections labeled by ChAT immunostaining show abnormal SAC somal lamination and disorganized dendrites in *Six3^Cre; Pten^cKO* retinas. (N) Quantification of SAC dendrite stratification using IPLaminator shows aberrant dendrite stratification in *Six3^Cre;Pten^cKO* compared to controls (WT: *n*=8; cHet: *n*=9; cKO: *n*=6) (*P*=0.0004; Kruskal-Wallis with Dunn's multiple comparisons test). (O-Q) P28 *ChAT^Cre;Pten^WT;Ai9*, *ChAT^Cre;Pten^cHet;Ai9* and *ChAT^Cre;Pten^cKO;Ai9* retina cross-sections with SAC somas and dendrites labeled using tdTomato. SAC somal lamination and dendrite organization are grossly normal, with two distinct bands in S2 and S4 in the IPL. (R) Quantification of SAC dendrite stratification shows no significant changes in *ChAT^Cre;Pten^cKO;Ai9* SACs relative to controls (WT: *n*=4; cHet: *n*=2; cKO: *n*=3) (*P*=0.4638; Kruskal-Wallis with Dunn's multiple comparisons test). Data are mean±s.e.m. Scale bars: 25 µm.

segregated synaptic inputs and outputs (Poleg-Polsky et al., 2018; Ding et al., 2016). Therefore, local increases in dendritic density could lead to a biased recruitment of specific pre- and post-synaptic partners.

### Dendrite branching phenotypes arise late in the development of $ChAT^{Cre};Pten^{cKO}$ SACs and persist into adulthood

SACs undergo extensive dendritic arborization during the first two postnatal weeks, increasing arbor territory and terminal branch number (Ing-Esteves and Lefebvre, 2024). To address when dendritic branching alterations arise in $ChAT^{Cre};Pten^{cKO}$ SACs, we examined SAC morphology prior to (P7) and after (P14) eye opening. We used immunohistochemistry to confirm that PTEN levels were significantly reduced in $ChAT^{Cre};Pten^{cKO}$ SACs at P7 (Fig. S3A-C). To label P7 SACs, we crossed the $ChAT^{Cre};Pten$ line with a $TIGRE-MORF$ ($Ai166$) reporter, which stochastically expresses EGFP in 1-5% of $Cre$-positive cells (Veldman et al., 2020). At P7, SAC dendrites are in a highly dynamic state, which is crucial for establishing their stereotyped radially symmetric morphology (Ing-Esteves and Lefebvre, 2024). Since there were no differences between $ChAT^{Cre};Pten^{WT}$ and $ChAT^{Cre};Pten^{cHet}$ SACs at P21, we included both genotypes as controls ($ChAT^{Cre};Pten^{Ctrl}$). We focused our analysis on GCL SACs, as the imaging resolution was better than INL SACs. Reconstruction and quantification of SACs identified no significant changes in total dendrite length, number of branch points, dendritic field area or soma size between $ChAT^{Cre};Pten^{Ctrl}$ and $ChAT^{Cre};Pten^{cKO}$ SACs at P7 (Fig. S4A-F). By P14, SACs have a much sparser dendritic arbor, with a morphology similar to their mature morphology. Using sparse viral labeling, we found that total dendritic length, branch number and dendritic field area remain unchanged in $ChAT^{Cre};Pten^{cKO}$ SACs at P14 compared to controls (Fig. S4G-K). However, we did detect a significant increase in soma size in $ChAT^{Cre};Pten^{cKO}$ SACs, suggesting that somal hypertrophy precedes changes in the dendritic arbor (Fig. S4L). Taken together, our results indicate that loss of PTEN from SACs drives excess dendritic branching between P14 and P21.

To assess whether the increased dendritic branching seen at P21 would resolve, persist or worsen in adulthood, we injected $AAV8$-$FLEx-tdTomato-CAAX$ at P28 and examined SACs at P60 (Fig. 4A,B). Similar to P21, adult (P60) $ChAT^{Cre};Pten^{cKO}$ SACs showed a near doubling of dendrite branching across their arbor despite no change in total dendrite length (Fig. 4D,E). Sholl analysis at P60 also largely recapitulated the phenotypes at P21, showing increased branch density in the outer 50% of the dendritic arbor (Fig. 4G). However, we did identify distinctions between P21 and P60; notably, $ChAT^{Cre};Pten^{Ctrl}$ SACs had 3-5 proximal dendrites of similar sizes, whereas $ChAT^{Cre};Pten^{cKO}$ SACs frequently had a prominent single hypertrophic dendrite (Fig. 4A′,B′). We defined any dendrite with a caliber greater than 1 μm as a 'hypertrophic dendrite' and found that these were present in 13/16 of $ChAT^{Cre};Pten^{cKO}$ SACs, compared with 1/12 in controls (Fig. 4C). $ChAT^{Cre};Pten^{cKO}$ SACs also had slightly smaller dendritic field areas compared to $ChAT^{Cre};Pten^{Ctrl}$ SACs (Fig. 4F). Despite these changes, SACs at P60 showed no changes in cell density, indicating that cell death was not occurring (Fig. S5A-C). These results show that the long-term loss of $Pten$ in SACs results in a persistent alteration in their dendritic arbor morphology.

### Loss of $Pten$ in SACs results in increased mTOR signaling over the course of development

PTEN serves as the primary negative regulator of the PI3K-AKT signaling pathway, which in turn activates mTOR signaling and inhibits GSK3β signaling (Fig. 5K) (Worby and Dixon, 2014). Both

mTOR and GSK3β alter the growth capacities of neurons, and are likely candidates to regulate SAC branching (Kosillo et al., 2022). We therefore examined how the loss of $Pten$ from SACs affects these pathways using an antibody to pS6 as a readout of mTOR activity (Cantrup et al., 2012) and a genetically-encoded β-catenin:GFP reporter ($TCF/Lef:H2B-GFP$) as a proxy for GSK3β signaling (Fig. 5A-H) (Ferrer-Vaquer et al., 2010). In the GCL of P28 $ChAT^{Cre};Pten^{cHet}$ retinal flat mounts, pS6 was undetectable in SACs, while it was present in a subset of RGCs. In contrast, all GCL SACs in $ChAT^{Cre};Pten^{cKO}$ retinas showed elevated pS6, indicating activation of mTOR signaling (Fig. 5I). Quantification of GFP signal in $ChAT^{Cre};Pten^{cHet};TCF/Lef:H2B-GFP$ and $ChAT^{Cre};Pten^{cKO};TCF/Lef:H2B-GFP$ retinas showed minimal fluorescence in SACs in both genotypes, suggesting that GSK3β signaling is unaffected by the absence of $Pten$ (Fig. 5J). Together, these results suggest that the morphological changes in $ChAT^{Cre};Pten^{cKO}$ SACs arise at least in part due to increased mTOR activity (Fig. 5K,L).

Since SACs did not show any changes in dendritic branching during the most dynamic time of dendritic growth (P7-P14), we assessed pS6 at these ages. At P7, $ChAT^{Cre};Pten^{cHet}$ SACs showed high levels of pS6, which was not further elevated in $ChAT^{Cre};Pten^{cKO}$ SACs (Fig. 6A,B,G,H). By P14, most SACs in $ChAT^{Cre};Pten^{cHet}$ mice had pS6 levels that were near background (Fig. 6C,C′,G,H), whereas pS6 was significantly increased in $ChAT^{Cre};Pten^{cKO}$ SACs (Fig. 6D,D′,G,H). Elevated pS6 levels were maintained in $ChAT^{Cre};Pten^{cKO}$ SACs at P60 (Fig. 6E-H). These results suggest that the lack of a dendritic branching phenotype in $ChAT^{Cre};Pten^{cKO}$ SACs at P7 may be because mTOR is already elevated at this age and loss of PTEN cannot drive further mTOR activity. In contrast, from P14 onwards mTOR activity has decreased in $ChAT^{Cre};Pten^{cHet}$ SACs, and deletion of $Pten$ results in persistently elevated mTOR activity, which maintains the branching and arborization process, leading to an increase in branch number by P21 and dendrite caliber by P60 (Fig. 6I).

To further test whether elevated mTOR was responsible for the altered dendrite branching in $ChAT^{Cre};Pten^{cKO}$ SACs, we tested whether treatment with Rapamycin could rescue this phenotype. We examined SACs in $ChAT^{Cre};Pten^{cHet}$ and $ChAT^{Cre};Pten^{cKO}$ mice following daily doses of Rapamycin (2.5 mg/kg, i.p.) between P14 and P28. Rapamycin treatment resulted in reduced body weight in both $ChAT^{Cre};Pten^{cHet}$ and $ChAT^{Cre};Pten^{cKO}$ mice compared to vehicle-treated mice, showing that this regimen was sufficient to reduce mTOR activity organism-wide (Fig. S6F,G). Rapamycin treatment also reduced the elevated pS6 in $ChAT^{Cre};Pten^{cKO}$ SACs back to control levels (Fig. S6A-E). Analysis of dendrite morphology showed that Rapamycin did not affect branching in $ChAT^{Cre};Pten^{cHet}$ SACs (Fig. S6H-N). Administration of Rapamycin in $ChAT^{Cre};Pten^{cKO}$ mice reduced SAC branching to a level that was not statistically different from $ChAT^{Cre};Pten^{cHet}$ SACs. This reduction did not reach statistical significance compared to vehicle treated $ChAT^{Cre};Pten^{cKO}$ SACs. Therefore, we interpret these results as Rapamycin causing a partial attenuation of $Pten$-induced hypertrophic branching and conclude that elevated mTOR signaling contributes in part to the branching phenotypes seen in $ChAT^{Cre};Pten^{cKO}$ SACs.

### SAC synaptic outputs and direction-selective circuit function are unaffected by loss of $Pten$

SACs have a highly compartmentalized synaptic organization, with presynaptic inputs from bipolar cells localized to the inner two-thirds of their dendritic arbor, and synaptic outputs localized to the outer one-third (Briggman et al., 2011). To

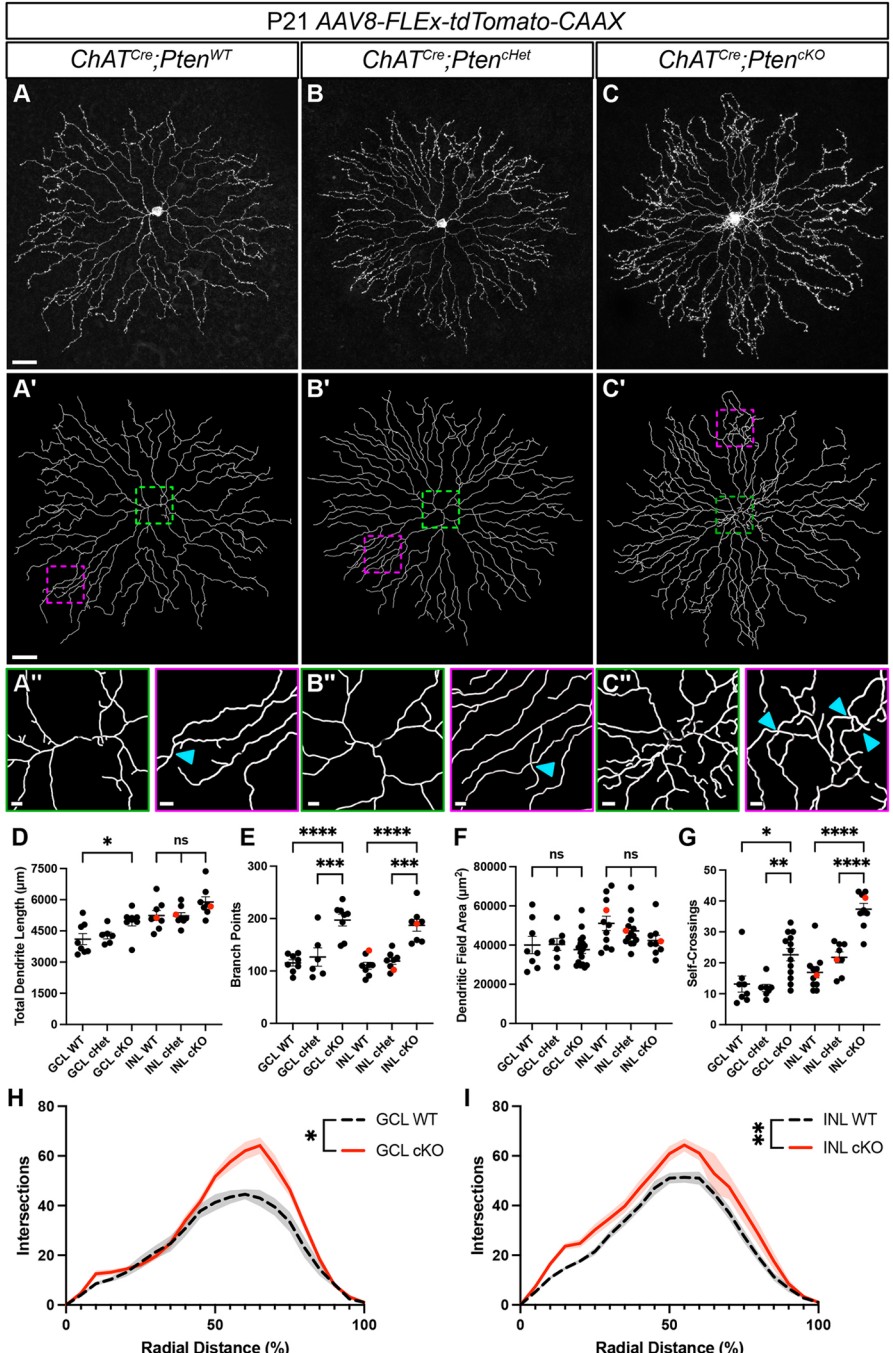

**Fig. 3.** *Pten*-deficient SACs have abnormal dendritic branching patterns. (A-C) SACs from P21 *ChAT^Cre^;Pten^WT^*, *ChAT^Cre^;Pten^cHet^* and *ChAT^Cre^;Pten^cKO^* retina flat mounts sparsely labeled with *AAV8-FLEx-tdTomato-CAAX*. Images show single SACs located in the inner nuclear layer (INL). (A′-C′) Imaris reconstructions of the SACs in A-C. (A″-C″) The increased branching on proximal dendrites (green outlined image) and crossovers (blue arrowheads) in distal dendrites (magenta outlined images). (D-G) Quantification of total dendrite length (GCL WT: *n*=8, 4106±264.9; GCL cHet: *n*=6, 4280±162.8; GCL cKO: *n*=8, 4953±214.4; INL WT: *n*=8, 5243±242.2; INL cHet: *n*=8, 5219±161.4; INL cKO: *n*=8, 5890±252.3) (GCL *P*=0.033; INL *P*=0.076; one-way ANOVA with post-hoc Tukey HSD test), number of branch points (GCL WT: *n*=8, 115.4±5.937; GCL cHet: *n*=6, 126.7±17.52; GCL cKO: *n*=8, 197.1±11.16; INL WT: *n*=8, 110.0±6.716; INL cHet: *n*=8, 118.9±5.962; INL cKO: *n*=8, 187.1±11.32) (GCL and INL *P*<0.0001; one-way ANOVA with post-hoc Tukey HSD test), dendritic field area (GCL WT: *n*=8, 40,012±4399; GCL cHet: *n*=7, 40,382±3074; GCL cKO: *n*=17, 37,723±1944; INL WT: *n*=11, 51,104±3668; INL cHet: *n*=14, 47,263 ±2269; INL cKO: *n*=9, 42,333±2714) (GCL *P*=0.756; INL *P*=0.152; one-way ANOVA with post-hoc Tukey HSD test) and dendrite branch self-crossings (GCL WT: *n*=8, 13.13±2.601; GCL cHet: *n*=7, 11.86±1.164; GCL cKO: *n*=17, 22.62±1.979; INL WT: *n*=11, 16.91±1.781; INL cHet: *n*=14, 21.75±1.770; INL cKO: *n*=9, 37.33±1.900) (GCL *P*=0.0015; INL *P*<0.0001; one-way ANOVA with post-hoc Tukey HSD test). A significant increase in the number of branch points and self-crossings is present in both ganglion cell layer (GCL) and inner nuclear layer (INL) SACs in *ChAT^Cre^;Pten^cKO^* retinas. Red dots indicate data from representative images. (H,I) Sholl analysis reveals differences in local density that differ between GCL and INL SACs in *ChAT^Cre^;Pten^cKO^* retinas (GCL WT: *n*=8, GCL cKO: *n*=8; INL WT: *n*=8, INL cKO: n=8) (GCL *P*=0.0312; INL *P*=0.0011; area under the curve analysis followed by a two-tailed unpaired Student's *t*-test). GCL cKO SACs show increased density near their terminal arbors, while INL cKO SACs show increased density throughout their arbor. Data are mean±s.e.m. and contain cells from at least three animals. Scale bars: 25 μm in A-C and A′-C′; 3 μm in A″-C″.

examine whether the loss of *Pten* affected the number or compartmentalization of SAC synapses, we injected *AAV1-FLEx-mGFP-2A-Synaptophysin-mRuby* to label their dendritic arbors with membrane-bound GFP and their synaptic outputs with synaptophysin (Syp) fused to mRuby (Beier et al., 2015; Koizumi et al., 2007) (Fig. 7A-B). Both *ChAT^Cre^;Pten^cHet^* and *ChAT^Cre^; Pten^cKO^* SACs showed robust localization of Syp:mRuby to the outer one-third of their dendritic arbor at P28 (Fig. 7A′-B′). We saw no differences in the number, volume and spatial distribution of Syp:mRuby puncta between *ChAT^Cre^;Pten^cKO^* SACs and controls (Fig. 7C-F). Therefore, even though SAC dendrite branching is dysregulated by P28 in *ChAT^Cre^;Pten^cKO^* SACs, synaptic outputs appear largely intact.

SACs have the conserved and well-characterized function of providing GABAergic inhibition and cholinergic excitation onto direction-selective ganglion cells (DSGCs) to tune direction selectivity (Ford and Feller, 2012; Pei et al., 2015). Genetic deletion of either *Sema6a* or γ-Pcdhs dramatically alters SAC morphology and degrades the direction selectivity of postsynaptic DSGCs (Sun et al., 2013; Kostadinov and Sanes, 2015). Therefore, we used multielectrode array (MEA) recordings to assess whether SAC-specific deletion of *Pten* influences downstream DSGCs. We isolated recordings from individual cells and computed their direction-selective index (DSI) in response to bars of light moving in 30° increments (Fig. 7G,H). A von Mises goodness of fit test was performed to determine if a cell matched established DSGC response properties (Yao et al., 2018). Cells with a DSI greater than 0.37 and a von Mises fit greater than 0.5 were classified as DSGCs. There was no significant difference in the distribution of DSIs of all RGCs between *ChAT^Cre^;Pten^cHet^* and *ChAT^Cre^;Pten^cKO^* retinas (Fig. 7I). The average DSI, von-Mises fit, average number of spikes per epoch, average spikes in preferred direction and tuning width in cells classified as DSGCs were unaffected in *ChAT^Cre^;Pten^cKO^* retinas compared to controls (Fig. 7J-N). Taken together, these data show that, while *Pten*-deficient SACs have significant morphological changes at P60, the function of the direction-selective circuit is unaffected.

## DISCUSSION

The process by which neurons develop stereotyped morphologies must be highly regulated to ensure consistency across each cell population. Here, we show that PTEN is required for SACs to adopt their precise branching patterns. By deleting *Pten* selectively from post-migratory SACs beginning at early postnatal ages, we were able to isolate the cell-autonomous function of PTEN and uncover its role in regulating dendrite branching at later stages of development. Mechanistically, this appears to result, in part, from elevated mTOR signaling, which normally decreases as SAC development progresses, but remains elevated in *Pten*-deficient SACs. Finally, we show that, despite altering the branching patterns of SAC dendrites, the loss of PTEN does not appear to disrupt the precise organization of synaptic outputs or the function of downstream retinal circuitry. Refining the cell-autonomous function of PTEN in SAC development.

The environment in which cells develop plays a crucial role in regulating their morphological features. Manipulating the pathways involved in regulating neuronal morphological development in intact preparations that maintain the extracellular environment can present challenges, as genetic deletions can lead to widespread anatomical changes that could affect neuronal morphology non-cell-autonomously. For example, pan-retinal *Pten* deletion broadly affects retinal progenitor proliferation, differentiation, somal lamination and dendrite stratification of multiple inner retinal neuron types (Cantrup

et al., 2012; Sakagami et al., 2012; Tachibana et al., 2016). The SAC somal organization and dendrite stratification phenotypes in these *Pten*-deficient retinas resemble defects seen in mice lacking specific transmembrane adhesion proteins important for SAC development, suggesting that PTEN could regulate the function of these proteins. Furthermore, deletion of *Pten* from retinal progenitors results in abnormal endocytic trafficking of cell-surface proteins and signaling molecules important for SAC migration, mosaic spacing and dendrite development (Touahri et al., 2024). However, it is equally possible that some of the SAC developmental phenotypes seen in pan-retinal *Pten* mutants are indirect and due to the profoundly disrupted extracellular environment.

To address this issue, we compared the effects of pan-retinal (*Six3^Cre^;Pten^cKO^*) and SAC-specific (*ChAT^Cre^;Pten^cKO^*) deletion of *Pten* side-by-side to disentangle its non-cell and cell-autonomous roles in SAC development. Like previous studies, we observed profound disruptions in SAC somal lamination and dendrite stratification after pan-retinal deletion of *Pten*. In contrast, when this deletion was restricted to post-migratory SACs, we observed normal mosaic spacing and dendrite stratification in the IPL, suggesting PTEN is not required for these processes. This could be due to the timing of *Pten* deletion; pan-retinal deletion of *Pten* occurs in retinal progenitors between E8.5 and E9.5, whereas recombination in *ChAT^Cre^;Pten^cKO^* mice begins at P1 as these cells are tangentially migrating to space their somas and beginning to stratify their dendrites in the nascent IPL (Liu and Cvekl, 2017; Marquardt et al., 2001; Rowan and Cepko, 2004; Ray et al., 2018). However, mice in which *Megf10* or *Plexa2* (*Plxna2*) are deleted using *ChAT^Cre^* still show defects in SAC mosaic arrangement and dendrite stratification, respectively (Ray et al., 2018; Sun et al., 2013). Therefore, the most likely explanation is that PTEN is not essential for the molecular pathways that regulate SAC somal positioning or dendrite stratification, while it is required for proper SAC dendrite branching. We also attempted to selectively delete *Pten* earlier from migrating SACs using *Megf10^Cre^* to examine SACs migration and mosaic spacing; however, these mice die immediately after birth.

### The role of PTEN in regulating neuronal morphology

PTEN has been extensively studied in the nervous system due to its identification as an autism risk gene (Garcia-Junco-Clemente and Golshani, 2014). In cultured hippocampal neurons, knockdown of *Pten* increases dendrite branching through the PI3K-AKT-mTOR pathway (Jaworski et al., 2005). *In vivo* deletion of *Pten* also causes generalized increases in dendritic growth and branching in dentate granule cells, cortical neurons and raphe serotonergic neurons (Santos et al., 2017; Gallent and Steward, 2018; Arafa et al., 2019; Chen et al., 2021; Getz et al., 2022; Kwon et al., 2006). However, in *Drosophila* dorsocentral neurons, RNAi knockdown of *Pten* primarily caused localized branching, as opposed to widespread neuronal hypertrophy, suggesting that the role of PTEN in regulating dendrite branching can differ depending on the neuronal subtype (Urwyler et al., 2019). Similarly, we found, while that *Pten*-deficient SACs displayed somal hypertrophy and increased dendrite branching, they maintained their overall dendritic arbor size. It is unclear why deletion of *Pten* from SACs does not increase dendritic arbor size, as it does in many other neuronal subtypes. Arbor size is tightly regulated in SACs, allowing them to create an even coverage factor across the retina. However, SACs do have the capacity to grow larger dendritic arbors, as deletion of the transmembrane gene *Amigo2* causes overall SAC dendritic arbor size to scale 50% larger, while branching is unaffected (Soto et al.,

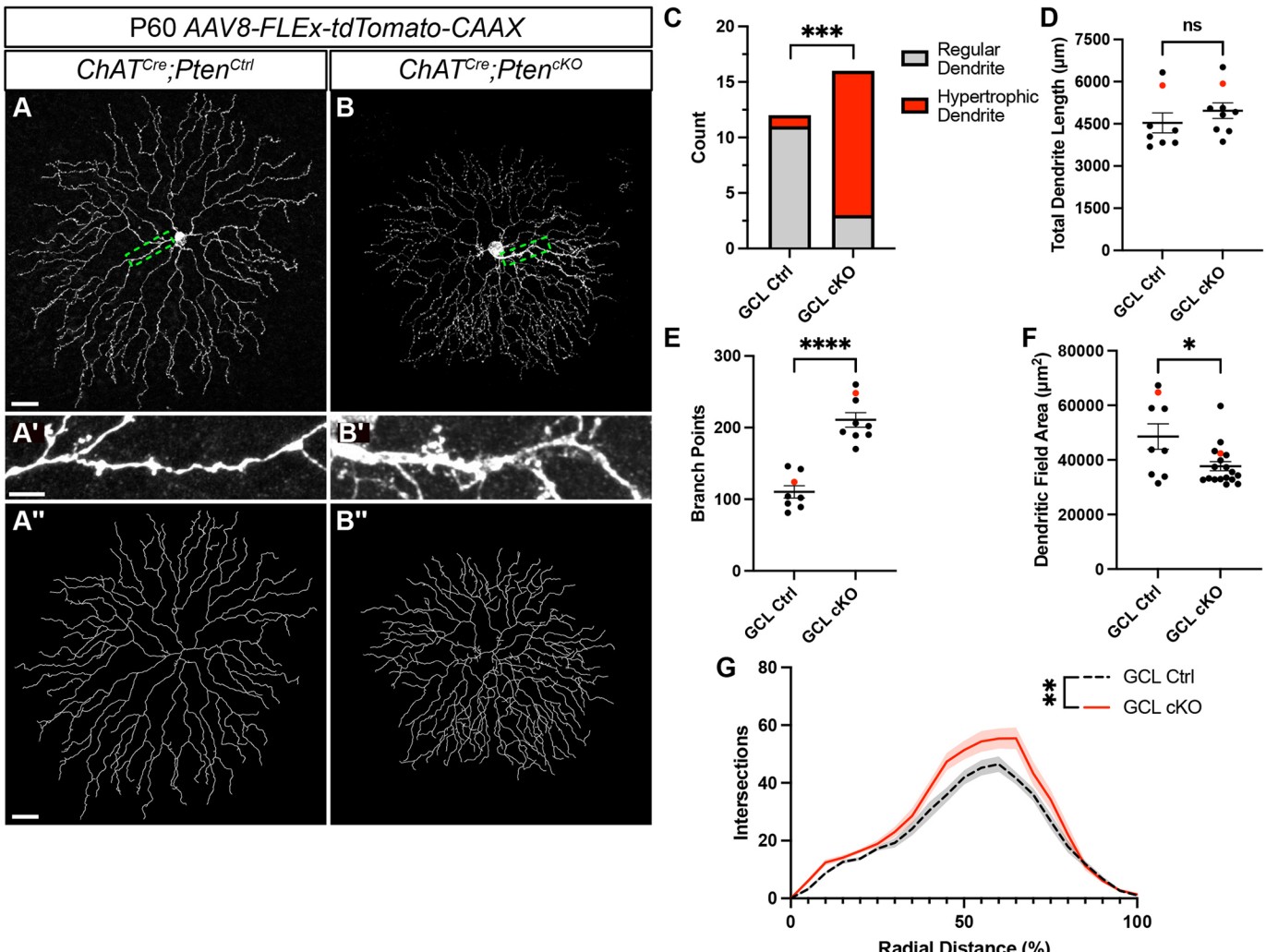

**Fig. 4. *Pten*-deficient SACs continue to show dendritic abnormalities at P60.** (A,B) P60 *ChAT^Cre^;Pten^Ctrl^* and *ChAT^Cre^;Pten^cKO^* SACs sparsely labeled by injection of *AAV8-FLEx-tdTomato-CAAX*. Images show single SACs located in the ganglion cell layer (GCL). (A′,B′) Enlargements of A and B highlighting that one of the dendrites frequently becomes hypertrophic in *ChAT^Cre^;Pten^cKO^* SACs. (A″,B″) Imaris reconstructions of SACs in A and B. (C) Quantification of the number of SACs containing a hypertrophic dendrite with a caliber greater than 1μm (8.33% in *Pten^Ctrl^* and 81.25% in *Pten^cKO^* SACs; P=0.0003 by Fisher's exact test). (D-F) Quantification shows that P60 cKO SACs have normal total dendritic length (Ctrl: n=8, 4535±353.8; cKO: n=9, 4968±277.5) (P=0.345; two-tailed unpaired Student's t-test), an increased number of branch points (Ctrl: n=8, 110.3±8.612; cKO: n=9, 210.8±10.21) (P<0.0001; two-tailed unpaired Student's t-test) and reduced dendritic field area (Ctrl: n=9, 48,556±4680; cKO: n=18, 37,693±1690) (P=0.013; two-tailed unpaired Student's t-test). Red dots indicate data from representative images. (G) Sholl analysis reveals increases in local dendrite branch density similar to the phenotype observed at P21 (Ctrl: n=8, cKO: n=9) (P=0.0043; area under the curve analysis followed by a two-tailed unpaired Student's t-test). Data are mean±s.e.m. and contain cells from at least three animals. Scale bars: 25 μm in A,B,A″,B″; 2 μm in A′,B′.

2019). Based on the normal dendritic arbor size in *Pten*-deficient SACs, we can conclude that PTEN is not required downstream of AMIGO2. These findings reinforce the hypothesis that the molecular mechanisms that modulate SAC dendritic field size and dendritic arbor branching are discrete processes. *Pten* deletion from SACs also did not significantly affect dendrite lamination in the IPL, suggesting it is not essential for the transmembrane proteins shown to regulate this process (Sun et al., 2013; Prigge et al., 2023).

γ-Pcdhs, semaphorin 6A and/or plexin A2 are required cell-autonomously within SACs to regulate dendrite self-avoidance (Lefebvre et al., 2012; Ing-Esteves and Lefebvre, 2024; James et al., 2024; Sun et al., 2013). γ-Pcdh mutant SACs have a normal number of terminal branches and overall arbor size, while *Sema6a/Plexa2* mutant SACs show reduced branching and arbor size. The dendrite morphology phenotypes that arise from mutations in these genes

are much more severe that those we observed following deletion of *Pten*. Therefore, while it is possible that PTEN can function downstream of γ-Pcdhs and semaphorin 6A and/or plexin A2, there are clearly additional signaling pathways required. For plexin A2, Rac GTPases are likely candidates, as SACs in mice with a point mutation that abolishes plexin A2 RasGAP activity show significant dendritic self-avoidance phenotypes (James et al., 2024).

## Dysregulation of the PI3K-AKT-mTOR pathway following *Pten* deletion in SACs

Deletion of *Pten* from SACs is likely to have multiple effects, including dysregulation of the PI3K-AKT-mTOR pathway (Worby and Dixon, 2014). mTOR activation is a powerful enhancer of neuronal growth, and deletion of its upstream inhibitor *Tsc1* causes dendritic hypertrophy that largely recapitulates *Pten* deficiency in

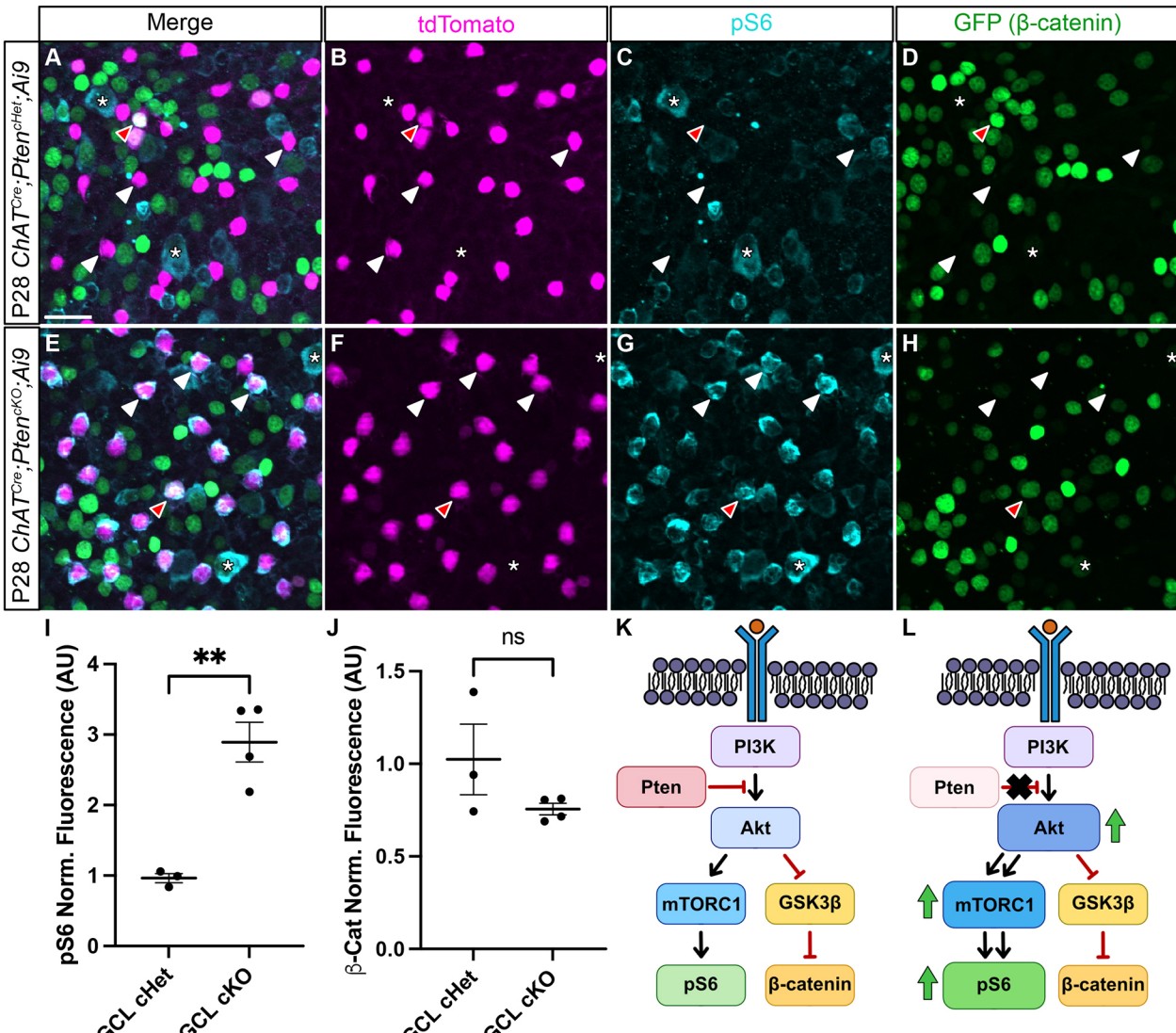

**Fig. 5. Deletion of *Pten* from SACs upregulates mTOR but not GSK3β signaling.** (A-H) Flat-mount preparations of P28 *ChAT^Cre^;Pten^cHet^;Ai9;Tcf/Lef:H2B-GFP* and *ChAT^Cre^;Pten^cKO^;Ai9;Tcf/Lef:H2B-GFP* retinas immunostained for tdTomato (magenta, B,F), pS6 (teal, C,G) and β-catenin reporter *Tcf/Lef:H2B-GFP* (green, D,H). White arrowheads highlight SAC cell bodies in the ganglion cell layer (GCL) in both *Pten* controls (A-D) and cKOs (E-H). Red arrowheads indicate a SAC with elevated levels of β-catenin reporter signal. Asterisks indicate retinal ganglion cells in both the control and cKO retinas that show elevated levels of pS6. (I,J) Quantification of pS6 and β-catenin fluorescence intensity shows a significant increase in pS6 levels (cHet: *n*=3, 0.9636±0.06534; cKO: *n*=4, 2.892±0.2812) (*P*=0.009; two-tailed unpaired Student's *t*-test) but no change in β-catenin levels (cHet: *n*=3, 1.024±0.1907; cKO: *n*=4, 0.7556±0.03099) (*P*=0.312; two-tailed unpaired Student's *t*-test) in cKO SACs. (K,L) Schematics showing a simplified view of the PI3K-AKT pathway. In the absence of *Pten* in SACs, AKT appears to increase mTOR activity, as measured by pS6 levels, while GSK3β signaling, as measured by β-catenin activity, remains unchanged. Data are mean±s.e.m. Scale bar: 25 μm.

cortical and olfactory bulb neurons (Kosillo et al., 2022; Feliciano et al., 2011). Conversely, inhibition of mTOR via rapamycin is sufficient to rescue overgrowth phenotypes from *Pten* loss in dentate granule neurons (Tariq et al., 2022). Our results show that pS6 levels are elevated following the deletion of *Pten*, suggesting that increased mTOR activity plays a role in regulating dendrite branching in SACs. In agreement with these results, reducing mTOR activity with Rapamycin treatment partially attenuated the increased branching phenotype in *Pten*-deficient SACs. Previous work has shown that both mTORC1 and mTORC2 activity can differentially influence dendrite morphology *in vivo*. Genetic manipulations that specifically inhibit mTORC1 in midbrain dopaminergic neurons result in reduced dendritic complexity and dendrite length, whereas inhibition of mTORC2 only affected proximal dendrite complexity, and had no

effect on the number of primary dendrites or total dendritic length (Kosillo et al., 2022; Feliciano et al., 2011). While Rapamycin is an acute and direct inhibitor of the mTORC1 complex, chronic Rapamycin treatment can result in indirect inhibition of mTORC2. Therefore, we cannot distinguish whether mTORC1 or mTORC2 regulate different aspects of dendrite branching in SACs at this time, and future studies will be needed to disentangle these pathways. GSK3β, another effector downstream of the PI3K-AKT pathway, can also regulate neurite outgrowth by modulating microtubule stabilization and polymerization (Zhou et al., 2004; Ziak et al., 2024). Our results using a genetic reporter of β-catenin activity as readout of GSK3β function detected no differences between control and *Pten*-deficient SACs, suggesting that this pathway does not play a major role in regulating dendrite branching in SACs.

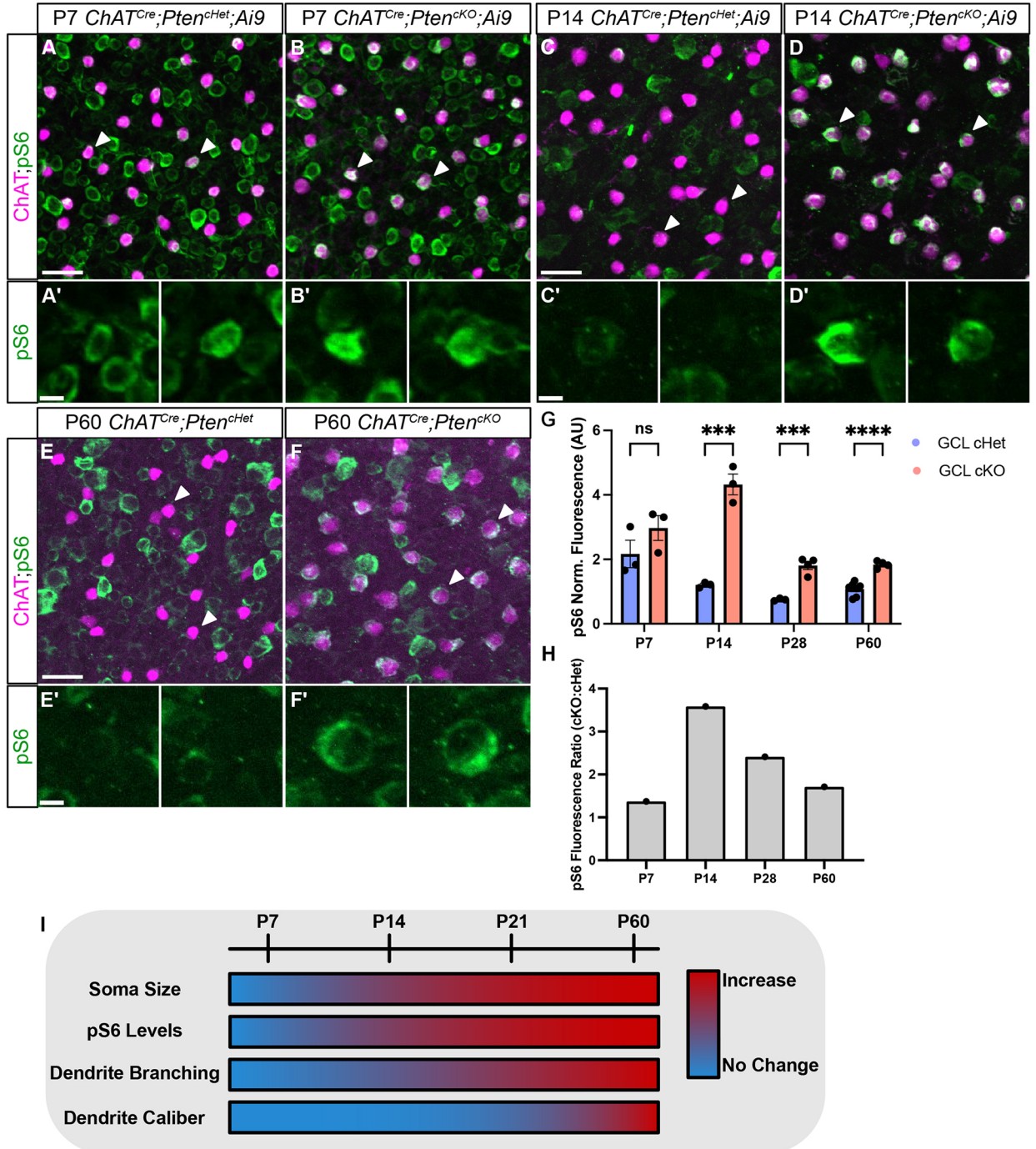

**Fig. 6. Increased pS6 precedes dendrite branching phenotypes in developing SACs.** (A-F) Retinal flat mounts of *ChAT^Cre^;Pten^cHet^* and *ChAT^Cre^;Pten^cKO^* SACs immunostained for ChAT (magenta) and pS6 (green) at P7 (A,B), P14 (C,D), and P60 (E,F). White arrowheads indicate SAC somas. (A′,B′) Higher magnifications P7 SAC somas indicated by arrowheads show high levels of pS6 in both *ChAT^Cre^;Pten^cHet^* and *ChAT^Cre^;Pten^cKO^* retinas. (C′,D′) At P14, pS6 levels are diminished in *ChAT^Cre^;Pten^cHet^* SACs (C′) but elevated in *ChAT^Cre^;Pten^cKO^* SACs (D′). (E′,F′) Higher magnifications of SAC somas show that pS6 remains elevated in *ChAT^Cre^;Pten^cKO^* SACs (F′) relative to controls (E′) at P60. (G,H) Quantification at P7, P14, P28 and P60 shows that pS6 levels are initially high in SACs at P7 (cHet: *n*=3, 2.168±0.4260; cKO: *n*=3, 2.971±0.3848) (*P*=0.2346) and decrease at later time points in control SACs, while pS6 levels remain significantly elevated in *ChAT^Cre^;Pten^cKO^* SACs relative to controls at P14 (cHet: *n*=3, 1.206±0.04906; cKO: *n*=3, 4.325±0.3233) (*P*=0.0007; two-tailed unpaired Student's *t*-test), P28 (cHet: *n*=5, 0.8165±0.04913; cKO: *n*=8, 2.351±0.2493) (*P*=0.0006; two-tailed unpaired Student's *t*-test) and P60 (cHet: *n*=7, 1.078±0.07916; cKO: *n*=4, 1.839±0.05409) (*P*<0.0001; two-tailed unpaired Student's *t*-test). (I) Summary of cellular phenotypes in *ChAT^Cre^;Pten^cKO^* SACs over the course of development and maturation. Increases in soma size and pS6 levels become apparent by P14, increased dendritic branching by P21, and localized increases in dendrite caliber are seen at P60. Scale bars: 25 µm in A,C,E; 5 µm in A′,C′,E′.

How does the loss of *Pten* in SACs result in increased dendritic branching? Elegant live imaging studies show that most of the SAC dendritic growth occurs between P4 and P14 (Ing-Esteves and Lefebvre, 2024). During the early part of this phase, dendrites contain many self-contacting interstitial protrusions that are highly dynamic and largely prune away by P14. These results suggest that

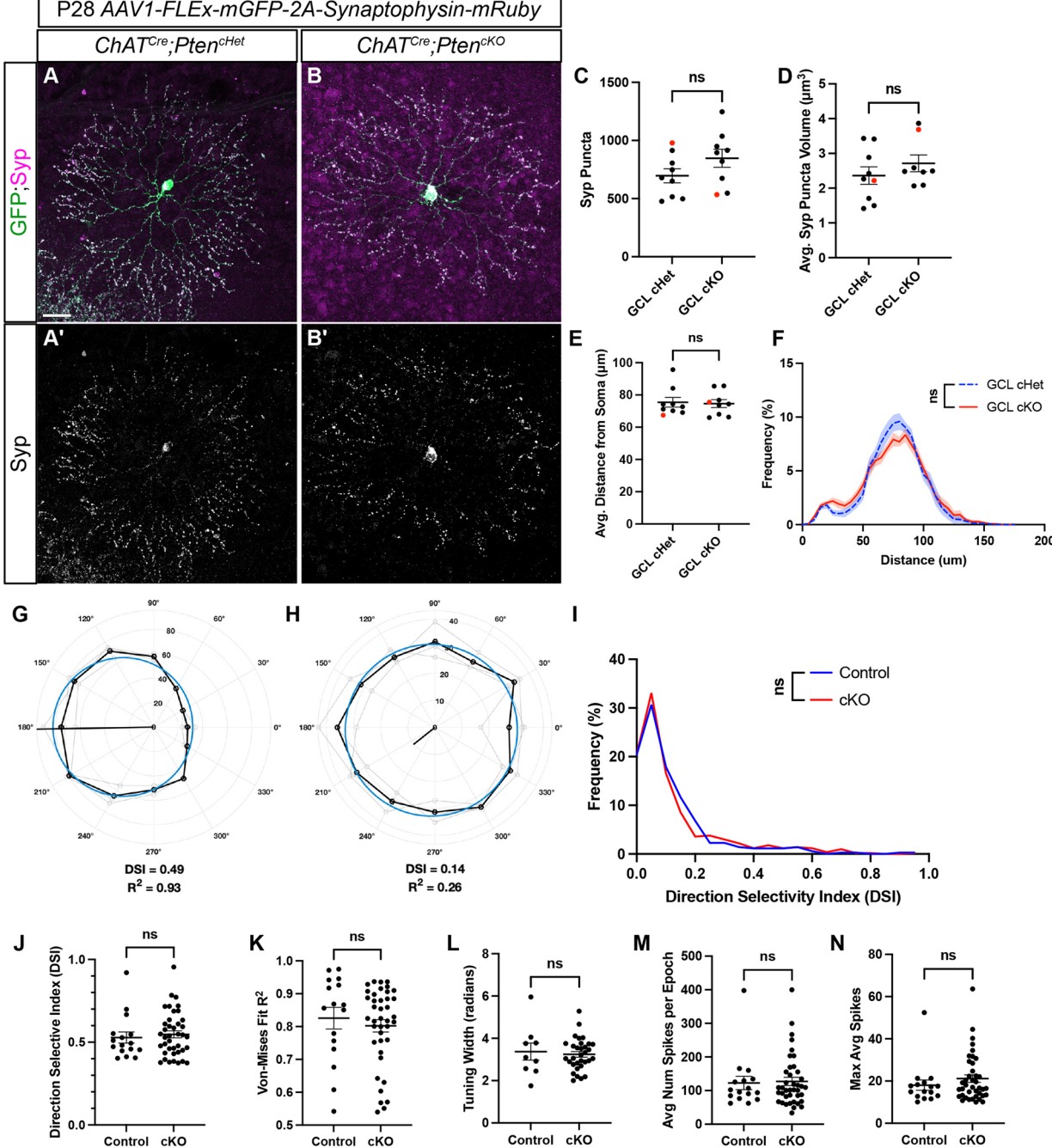

**Fig. 7. Loss of *Pten* does not affect SAC synaptic outputs or alter retinal responses to directional stimuli.** (A,B) P28 *ChAT^Cre^;Pten^Ctrl^* and *ChAT^Cre^; Pten^cKO^* retinas injected with *AAV1-FLEx-mGFP-2A-Synaptophysin-mRuby* to label SAC dendrites with membrane-bound GFP and synaptic release sites with synaptophysin (Syp) fused to mRuby. (A′,B′) Syp:mRuby shows highly compartmentalized localization to the outer one-third of SAC dendritic arbors in both *ChAT^Cre^;Pten^cHet^* and *ChAT^Cre^ Pten^cKO^* SACs. Scale bar: 25 μm. (C,D) Quantification of the number (cHet: *n*=9, 696.3±60.60; cKO: *n*=9, 846.8±77.48) (*P*=0.146) and volume (cHet: *n*=9, 2.362±0.2537; cKO: *n*=8, 2.716±0.2415) (*P*=0.332) of Syp:mRuby puncta shows no significant differences between *ChAT^Cre^;Pten^cHet^* and *ChAT^Cre^ Pten^cKO^* SACs (*P*=0.146 and *P*=0.332; two-tailed unpaired Student's *t*-test). Data are mean±s.e.m. (E,F) Quantification of the distribution of Syp:mRuby puncta reveals no significant changes in the average distance from the soma (cHet: *n*=9, 75.46±2.994; cKO: *n*=9, 74.66±2.452) (*P*=0.838; two-tailed unpaired Student's *t*-test) or the distribution along SAC dendrites (*P*>0.999; area under the curve analysis followed by a two-tailed unpaired Student's *t*-test) in *ChAT^Cre^;Pten^cHet^* and *ChAT^Cre^ Pten^cKO^* SACs. Data are mean±s.e.m. and contain cells from at least three animals. Data are mean±s.e.m. (G,H) Example polar plots of directional responses from individual RGCs following MEA recording of *ChAT^Cre^;Pten^cHet^* and *ChAT^Cre^ Pten^cKO^* retinas. Direction selective (DS) cells were identified based on their direction selective index (DSI) and their goodness of fit for the von Mises distribution. The black trace indicates the cells responses to light stimuli in different directions, while the blue circle represents the von Mises fit. (G) An example of a direction-selective response; (H) a direction-insensitive response. (I) Distribution of the DSI of all detected cells from MEA recordings shows that most cells fall below the DSI threshold for a DS cell (0.37), but a small population are direction selective (*P*=0.9999). (J-N) Quantification of DSI (J; *P*=0.632), von Mises fit (K; *P*=0.5179), tuning width (L; *P*=0.6965), average number of spikes per epoch (M; *P*=0.8411) and average spikes (N; *P*=0.3397) in a preferred direction from DS cells; no significant differences in the response properties of DS cells were detected between *ChAT^Cre^;Pten^cHet^* and *ChAT^Cre^ Pten^cKO^* retinas (two-tailed unpaired Student's *t*-test). Data are mean±s.e.m.

there is a normal developmental window of dynamic dendritic growth and remodeling in SACs that is largely completed by P14. While we did not conduct live imaging in our studies, our data show that loss of *Pten* predominantly affects SAC dendrite branching after P14, suggesting that PTEN-regulated pathways may be involved in the closing of this developmental window. At P7, when SACs are undergoing extensive dendritic growth, high levels of pS6 were seen in control SACs. Deletion of *Pten* had no effect on pS6, somal hypertrophy or dendrite branching at P7, suggesting that mTOR activity may already be near maximal levels at this age. By P14, we saw decreased levels of pS6 in control SACs, suggesting that the decline in mTOR activity normally coincides with when dendritic arbor growth and branching begins to slow. In contrast, *Pten*-deficient SACs show elevated pS6 and somal hypertrophy at this age, yet dendrite branching is unaffected. It is not until P21 and later that *Pten*-deficient SACs show increased dendrite branching. This suggests that loss of PTEN does not generally affect the major phase of developmental dendrite growth in SACs, but prolongs it beyond its normal plateau, resulting in excessive branching. The presence of excessive dendrite branches at P60 suggests that these branches are not unstable dynamic projections, but rather are persistent branches. Whether these aberrant branches are functionally integrated into retinal circuitry is unclear.

At P60, SACs in *ChAT^Cre^;Pten^cKO^* retinas began to show a decrease in dendritic arbor size, suggesting that prolonged deletion of *Pten* could have adverse effects in neurons. In Purkinje neurons, loss of *Pten* results in the eventual apoptotic death of these neurons, beginning around 6 months of age (Cupolillo et al., 2016). We did not observe any loss of SACs in *ChAT^Cre^;Pten^cKO^* retinas at P60, and did not examine later ages as these mice eventually develop facial tumors (Meyer Zu Reckendorf et al., 2022). However, it is important to consider the long-term consequences of *Pten* deletion in neurons, as it is widely studied for its ability to facilitate axon regeneration after injury (Park et al., 2008; Christie et al., 2010; Liu et al., 2010; Sun et al., 2011). While these studies rarely report the effect of *Pten* deletion on the dendrites of these cells, a recent report shows that RGC dendrites rapidly retract in response to axonal injury, and co-deletion of *Pten* and *Socs3* exacerbates this effect (Santos et al., 2025 preprint). Our results suggest that the long-term deletion of *Pten* in neurons can cause deleterious changes in dendritic architecture.

### Functional consequences of *Pten* loss in SACs

At a circuit function level, SACs are crucial for the directional tuning of downstream DSGCs (Yoshida et al., 2001; Amthor et al, 2002; Vlasits et al., 2014). SACs themselves display intrinsic direction selectivity, responding preferentially to centrifugal motion moving from the soma to the dendritic tips (Euler et al., 2002; Lee and Zhou, 2006). Mutations that disrupt the radial morphology of SAC dendritic arbors result in defective direction selectivity (Kostadinov and Sanes, 2015; Morrie and Feller, 2018; Sun et al., 2013). Loss of *Pten* in other neurons can dramatically affect their circuit function; hippocampal dentate granule neurons and serotonergic raphe neurons lacking *Pten* are hyperactive due to changes in their intrinsic excitability, and have an increased number of excitatory inputs, whereas *Pten*-deficient Purkinje neurons show reduced excitability (Luikart et al., 2011; Williams et al., 2015; Santos et al., 2017; Chen et al., 2021; Cupolillo et al., 2016). It is therefore somewhat surprising that, despite the abnormal dendritic branching in SACs, direction selectively appeared intact in *ChAT^Cre^;Pten^cKO^* retinas. We also observed no changes in the localization, density or size of SAC synaptic outputs in *ChAT^Cre^; Pten^cKO^* retinas. While the molecular mechanisms that underlie the

spatial segregation of synaptic inputs and outputs in SACs remain unknown, *Pten* signaling is apparently not required.

### Study limitations

While we have refined the role of PTEN in regulating the morphology of SACs, there are some limitations to our study. Our data showing increased pS6 in *Pten*-deficient SACs from P14 onwards suggest a role for mTOR signaling in mediating the increased dendrite branching, and this is supported by data showing that daily administration of Rapamycin partially attenuates the branching phenotypes. However, we cannot rule out a role for other signaling pathways downstream of Pten. We attempted to examine the role of mTOR signaling directly by generating both *ChAT^Cre^; Tsc1^cKO^* and *ChAT^Cre^;Tsc2^cKO^* mice. However, these mutant mice showed motor deficits and failed to thrive, resulting in death around P14, precluding analysis of SAC morphology. We presume that this early lethality is the result of motor neuron loss or dysfunction, as *ChAT^Cre^* also drives recombination in all cholinergic motor neurons. It is not clear why the phenotypes in the *Tsc1^cKO^* and *Tsc2^cKO^* mice are more severe than the *Pten^cKO^* mice. Future experiments using conditional deletion of raptor (*Rptor*) or *Rictor* could allow us to directly test the specific roles for mTORC1 and mTORC2, respectively, in regulating SAC morphology. Our data also showed that the branching phenotypes we observed in *Pten*-deficient SACs did not affect the overall function of the direction-selective circuit based on multielectrode array recordings. We did not examine the intrinsic physiological properties of *Pten*-deficient SACs directly, so it is possible that there are subtle cell-autonomous functional differences that would require direct analysis of their physiological properties. Ultimately, additional work will be needed to further understand the role that PTEN signaling plays in SAC development and function.

## MATERIALS AND METHODS
### Animals and animal procedures
All animal procedures were approved by the Oregon Health and Science Institutional Animal Care and Use Committee (IACUC). The following mouse lines were used: *ChAT^Cre^* (Rossi et al., 2011), *Six3^Cre^* (Furuta et al., 2000), *Pten^flox^* (Backman et al., 2001), *Ai9* (Madisen et al., 2010), *TIGRE-MORF/Ai166* (Veldman et al., 2020) and *TCF/Lef:H2B/GFP* (Ferrer-Vaquer et al., 2010) (see Table S1). All lines were maintained on a *C57BL/6J* background. Mice of both sexes were used for experiments. Mice were genotyped using the oligonucleotides listed in Table S1. Intravitreal injections were performed on mice at P2 or P28. P2 mice were anesthetized through indirect contact with ice and brought back to body temperature through contact with a warm glove. Mice were injected with a 30 psi pulse for 30 ms. P28 mice were anesthetized using isoflurane at a flow rate of 3%, and maintained at a flow rate of 1.5%. After being deeply anesthetized, animals were placed on a Kopf stereotaxic injection rig (Model 1900 Stereotaxic Alignment system). A 0.5% proparacaine hydrochloride ophthalmic solution was applied as a topical anesthetic to the eye, followed by 1% tropicamide ophthalmic solution for dilation. Gentle-eye lubricant was added to both eyes, and a microvascular clamp was used to push the globe outwards. Needles were pulled on a Sutter Instrument micropipette puller (P-97 Flaming/Brown Micropipette Puller) and beveled with sandpaper. Animals were injected with *AAV8-FLEx-tdTomato-CAAX* at a titer of $2.6\times10^{11}$ or with *AAV1-FLEx-mGFP-2A-Synaptophysin-mRuby* at a titer of $1.8\times10^{12}$ (Beier et al., 2015) (see Table S1).

### Tissue processing and immunohistochemistry
Mouse eyes were enucleated and drop fixed in 4% EM-grade PFA for 30 min. For *TIGRE-MORF* mice, retinas were instead fixed with a solution of 9% glyoxal 8% acetic acid at pH 4.0, as this improved resolution of the Tigre-GFP signal. After fixation, eyes were washed in PBS. Eyes were then

pierced with a 30 G ½ inch needle and the cornea was cut away with microdissection scissors.

For retinal flat mounts, the retina was isolated and then transferred to an Eppendorf tube, and blocked and permeabilized with a blocking solution (2% normal donkey serum, 0.2% Triton X-100 and 0.002% sodium azide) for 1 h. After blocking, retinas were stained with primary antibodies (see Table S1) and left shaking at 4°C for 3-4 days. Retinas were then washed overnight in PBS, stained with secondary antibodies (see Table S1) and left shaking at 4°C for 1 day. Four cuts were made into the retina to allow it to lay flat on glass microscope slides. Retinas were mounted with Fluoromount-G and sealed with nail polish.

For retinal cross-sections, the retina and lens were left in the eye cup and cryoprotected in 15% sucrose overnight. The next day, the lens was removed, eyes were placed in Neg-50 and the eyes were frozen in the eye cup in 2-methylbutane. Eyes were sectioned on a Leica cryostat (CM3050 S) at 20 μm. The edges of the slide were then coated with a hydrophobic barrier using an ImmEdge pen. The slide was washed with PBS to remove any remaining Neg-50 attached to the slide. Retinas were then stained with primary antibody at 4°C overnight. The next day, retinas were washed with PBS three times for 15 min. Secondary antibody was applied and the retinas were left at room temperature for 2 h. Hoechst was then applied to the retinas for 10 min, followed by three 15-min PBS washes. Retinas were mounted with Fluoromount-G and sealed with nail polish. Antibody dilutions were the same as in flat mounts. Details of reagents used can be found in Table S1.

### Fluorescence image acquisition

Retinal sections were imaged on a Zeiss Axio Imager M2 upright microscope equipped with an ApoTome2 using a 20× objective. Retinal flat mounts imaged for cell density, somal quantification, mosaic analysis, pS6 and β-catenin quantification were also imaged with these settings. Retinal flat mounts imaged for single cell morphology and synaptophysin labeling were imaged on a Zeiss LSM 900 confocal microscope using a 40× water objective with NA=1.2. Images were acquired using the Zeiss Zen Imaging software for both microscopes.

### Multi-electrode array recordings

Tissue from mouse retinae was placed RGC side down on a 3Brain Accura HD-MEA connected to a BiocCAM DupleX recording system (3Brain AG, Wädenswil, Switzerland). The Accura HD-MEA contains 4096 electrodes in a 3.8×3.8 mm area, where each electrode is 21 μm×21 μm spaced 60 μm apart. The internal diameter of the reservoir is 25 mm, with a 7 mm height. Retinae were dissected off the choroid, and the vitreous was then carefully dissected from the tissue prior to mounting the tissue photoreceptor-side down on Millicell polytetrafluoroethylene membrane cell culture insert (Millipore Sigma, PICM0RG50). To adhere the retina to the membrane, the Ames medium was removed from the insert and gentle suction was applied using a gas stone connected to a vacuum chamber with filter paper between the stone and the membrane. The insert was then placed back into Ames medium and the membrane trimmed to outside the edges of the retinae. The retinae and/or insert was placed RGC-side down on the MEA surface in Ames medium and the Ames medium was removed from the reservoir to facilitate connectivity with the electrodes. A platinum harp was placed over the insert to hold the retinae in place, the reservoir was filled with Ames, the MEA was transferred to the BioCAM DupleX and the reservoir was continuously superfused with Ames medium at 32°C. Details of reagents used can be found in Table S1.

Visual stimuli were generated using custom software generated in Python (PyStim; https://github.com/SivyerLab/pystim) and presented on a LightCrafter 4500 projector (Texas Instruments) modified by removing the focusing optics. Projector light was captured with a TV lens, passed through neutral density filters and an Olympus MVX10 fluorescence microscope system with a MVPLAPO 2XC (0.5 numerical aperture) objective. A 1.5 mm coverslip was mounted onto the reservoir to reduce diffraction caused by the air/Ames interface. Visual stimuli presented used either the green (LE CG Q9WP; 520 nm peak) or blue (LE B Q9WP; 455 nm peak) LEDs and were presented full field at a range of light intensities between 6.2e11 photons/cm2/s and 8.9e13 photons/cm2/s. Stimuli consisted of green full field chirp responses used to isolate rod-

and cone-mediated inputs to RGCs, ON and OFF responses, and temporal and spatial frequency tuning properties. Moving stimuli consisted of gratings moving in 12 directions for 3 s in each direction.

### Quantification and statistical analysis

#### SAC population analyses: cell counting, soma size, mosaic spacing, and fluorescence signal analysis of pS6, β-catenin and PTEN

Retina flat mounts were immunostained for either tdTomato or ChAT. Three regions were imaged per retina per animal. GCL and INL images were captured from the same regions of the retina, avoiding the center and far periphery. Every $Pten^{cKO}$ animal included had at least one $Pten^{cHet}$ littermate control. The tdTomato or ChAT channel was binarized in ImageJ using either the Otsu or Huang thresholding algorithm and the 'Analyze Particles' function of ImageJ was used to perform cell counts. As part of this process, we generated individual regions of interest (ROIs) for every cell, an ROI consisting of every cell and an inverse ROI consisting of the background.

As part of cell density quantification, the size of the particles and the $xy$ coordinates of the center of SAC somas were collected for soma size and mosaic spacing analysis. These $xy$ coordinates were then entered into WinDRP, a program designed to calculate the regularity of cells using a density recovery profile (Rodieck, 1991). From this, a regularity index ratio was calculated, which defines how mosaically spaced a population of cells are when compared with a random distribution.

The same image acquisition and ImageJ pipeline for cell counting was used to analyze pS6 and β-catenin fluorescence. The 'Analyze Particles' function of ImageJ was used to obtain a ROI for the SACs within an image and the inverse ROI representing background signal. The pS6 and β-catenin fluorescent signals were then measured in both these ROIs. The SAC ROI was then normalized to the background ROI to obtain a normalized fluorescence intensity signal. PTEN fluorescence at P7 was quantified by drawing ROIs around individual SACs using the ChAT signal to define the soma, then quantifying the total PTEN signal within that area. Background levels were calculated by quantifying signal in non-SAC areas devoid of PTEN, which was subtracted from the PTEN quantification from each cell. This approach was also used to quantify pS6 signal for individual SACs in Rapamycin dosing experiments.

#### IPLaminator analysis

Lamination was quantified in P28 retina cross-sections stained for tdTomato and DAPI using IPLaminator. IPLaminator is an ImageJ plugin designed to bin and quantify retinal lamination (Li et al., 2016). DAPI was used to define the area of the IPL, then tdTomato fluorescence was measured by IPLaminator. For every image, the IPL was divided into 20 equal sections and measured along the depth of the IPL to normalize any variance in IPL thickness.

#### Imaris reconstructions and morphometric analysis

Individual SACs were collected from central, medial and peripheral areas of the retina, and were excluded if they were within 100 μm of the optic nerve head or edge of the retina. SACs from all areas of the retina were included in this analysis. SACs were reconstructed manually using the Imaris Filaments module. Briefly, the soma was assigned as a dendrite beginning point and dendrites were traced from there using the AutoPath tool. From these reconstructions, data including total dendrite length, number of branch points, branch level and Sholl intersections were collected. Sholl data was generated from Imaris at 1μm intervals. Sholl data was normalized to the radial distance of the cell by averaging the number of Sholl intersections along every 10% of the radial distance. Dendrite field area was measured in ImageJ by taking the convex hull of the fluorescent area covered by a SAC. The number of self-crossings at P21 was manually counted in ImageJ by counting dendrite branch intersections in single z-planes across a z-stack. Dendrite caliber was measure in ImageJ by drawing a line perpendicular to the dendrite and measuring its length. Any dendrite greater than 1 μm in caliber was considered a hypertrophic dendrite.

#### Rapamycin treatment

Rapamycin was first dissolved into a stock solution at 50 mg/ml in ethanol and stored at −20°C. Rapamycin was then made up in a solution of 5% PEG-

400 and 5% Tween-80 in sterile saline at a concentration of 0.25 mg/ml. The vehicle solution was 5% PEG-400 and 5% Tween-80 in sterile saline. These solutions were used for 1 week and injected intraperitoneally into mice at a concentration of 10 μl/g body weight (2.5 mg/kg Rapamycin). Mice were injected daily from P14 to P28, and were taken for morphological analysis at P28. Details of reagents used can be found in Table S1.

### Synaptophysin analysis

Synaptophysin puncta were quantified in Imaris using the Surfaces module. Briefly, surfaces were generated for all putative synaptophysin puncta, which were then filtered based on the fluorescence of the membrane-bound GFP signal to exclude noise outside the SAC. Puncta smaller than 0.5 μm were also excluded. The number of puncta, size of the puncta and distance from the soma of every puncta was obtained through Imaris.

### Spike sorting

Herding Spikes 2 (HS2), via SpikeInterface[2] Python framework, was used for spike detection and sorting. HS2 uses a mixed approach of spike spatial and prominent waveform features combined with a mean shift clustering algorithm to identify individual cells and their corresponding spikes on the array. The spike sorting scripts were executed on a high-performance computing cluster (exacloud). The HS2 parameters used are provided in Table 1.

### DSGC classification and statistical analysis

Post spike sort, subsequent analysis in MATLAB R2023a (Mathworks) was performed for further cell classification based on light responses. Responses from individual units were assessed for each presented direction of light. Units without a minimum of 400 total spikes were filtered out. Remaining units had their direction selectivity $\left(\frac{Preferred\ Direction - Null\ Direction}{Preferred\ Direction + Null\ Direction}\right)$ and von Mises fit calculated (Yao et al., 2018). Units with a DSI greater than 0.37

and a von Mises fit greater than 0.5 were classified as putative DSGCs. Additional filtering was then performed to make sure the units had a minimum of 10 average spikes in their preferred direction across the epochs. Once DSGCs were determined, the average number of spikes per stimulus window (epoch) and the average number of spikes in their preferred direction were measured. Tuning width was calculated by using the full width at half maximum (FWHM) (Yao et al., 2018).

### Statistics

For each experiment and time point, a minimum of three mice per condition were analyzed. For single cell morphological analysis, a minimum of six cells were analyzed, with most conditions having at least eight cells. For all datasets, the variance was reported as mean±s.e.m. For analysis between two groups, an unpaired two-tailed Student's t-test was performed. For analysis between three groups, an ANOVA with Tukey's multiple comparison was performed. For comparisons of distributions, a Kruskal–Wallis test was performed. For Sholl and branch level analyses, an area under the curve (AUC) analysis was performed, where AUC statistics (mean, s.e.m. and $n$) were computed, then analyzed via an unpaired two-tailed Student's t-test. For comparison of the ratio of SACs with a hypertrophic dendrite, Fisher's exact test was used. All statistical tests were performed in GraphPad Prism 9.

### Acknowledgements

We thank the editors and reviewers for their helpful comments. We acknowledge expert technical assistance by staff in the OHSU Advanced Light Microscopy shared resource. We also thank John Brigande, Paul Barnes and members of the Wright lab for helpful suggestions over the course of the study.

### Competing interests

The authors declare no competing or financial interests.

### Author contributions

Conceptualization: T.W.B., K.M.W.; Data curation: T.W.B.; Formal analysis: T.W.B.; Funding acquisition: K.M.W.; Investigation: T.W.B.; Methodology: T.W.B., N.L., J.L., B.S.; Project administration: K.M.W.; Supervision: B.S., K.M.W.; Writing – original draft: T.W.B.; Writing – review & editing: T.W.B., N.L., J.L., B.S., K.M.W.

### Funding

This work was funded by grants from the National Eye Institute (R01EY032057 to K.M.W. and R01EY032564 to B.S.). T.W.B. was supported by training grants from the National Institute of Neurological Disorders and Stroke (T32NS007466) and the National Eye Institute (T32EY023211). Open Access funding provided by Oregon Heath & Science University. Deposited in PMC for immediate release.

### Data and resource availability

All relevant data and details of resources can be found within the article and its supplementary information.

### Peer review history

The peer review history is available online at https://journals.biologists.com/dev/lookup/doi/10.1242/dev.204980.reviewer-comments.pdf

### Table 1. HS2 parameters

| Parameter | Value |
| --- | --- |
| mea_pitch | 60 |
| electrode_width | 21 |
| clustering_bandwidth | 20 |
| clustering_alpha | 5.5 |
| clustering_n_jobs | −1 |
| clustering_bin_seeding | true |
| clustering_min_bin_freq | 16 |
| clustering_subset | null |
| left_cutout_time | 0.3 |
| right_cutout_time | 1.8 |
| detect_threshold | 20 |
| probe_masked_channels | [] |
| probe_inner_radius | 100 |
| probe_neighbor_radius | 129 |
| probe_event_length | 0.26 |
| probe_peak_jitter | 0.2 |
| t_inc | 100,000 |
| num_com_centers | 1 |
| maa | 12 |
| ahpthr | 11 |
| out_file_name | 'HS2_detected' |
| decay_filtering | false |
| save_all | false |
| amp_evaluation_time | 0.4 |
| spk_evaluation_time | 1.0 |
| pca_ncomponents | 2 |
| pca_whiten | true |
| freq_min | 300.0 |
| freq_max | 6000.0 |
| filter | true |
| pre_scale | true |
| pre_scale_value | 20.0 |
| filter_duplicates | true |

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
