## [Peer Review File · Development (Cambridge, England)]

PTEN regulates starburst amacrine cell dendrite morphology during development

Teva W. Bracha, Nina Luong, Joseph Leffler, Benjamin Sivyer and Kevin M. Wright
DOI: 10.1242/dev.204980

Editor: Debra L Silver

Review timeline

Original submission:	28 May 2025
Editorial decision:	19 July 2025
First revision received:	3 March 2026
Editorial decision:	31 March 2026
Second revision received:	1 April 2026
Accepted:	5 April 2026

Original submission

First decision letter

MS ID#: dev.204980

MS TITLE: PTEN regulates starburst amacrine cell dendrite morphology during development

AUTHORS: Teva Bracha; Nina Luong; Joseph Leffler; Benjamin Sivyer; Kevin M. Wright

Dear Dr Wright,

I have now received all the referees' reports on the above manuscript, and have reached a decision. The referees' comments are appended below, or you can access them online: please go to:

As you will see, the referees express considerable interest in your work, but have some significant criticisms and recommend a substantial revision of your manuscript before we can consider publication. The concern is raised that some claims are not supported by data. Please address with inclusion of new data and in some cases, modified conclusions as well as clarifications of interpretations. Both reviewers highlight some functional experiments needed to bolster the mechanisms. Keeping in mind the scope feasible for a revision, it would be particularly important to provide support for the extent to which mTOR signaling contributes to branching phenotypes.

If you are able to revise the manuscript along the lines suggested, I will be happy to receive a revised version of the manuscript. Your revised paper will be re-reviewed by one or more of the original referees, and acceptance of your manuscript will depend on your addressing satisfactorily the reviewers' major concerns. Please also note that Development will normally permit only one round of major revision. If it would be helpful, you are welcome to contact us to discuss your revision in greater detail. Please send us a point-by-point response indicating your plans for addressing the referees' comments, and we will look over this and provide further guidance.

Please attend to all of the reviewers' comments and ensure that you clearly highlight all changes made in the revised manuscript. Please avoid using 'Tracked changes' in Word files as these are lost in PDF conversion. I should be grateful if you would also provide a point-by-point response detailing

how you have dealt with the points raised by the reviewers in the 'Response to Reviewers' box. If you do not agree with any of their criticisms or suggestions please explain clearly why this is so.

Reviewer 1

SUMMARY OF THE ADVANCE MADE IN THIS PAPER AND ITS POTENTIAL SIGNIFICANCE TO THE FIELD

This manuscript examines the role of PTEN on the postnatal development and maintenance of dendritic arborizations, with a focus on the cholinergic starburst amacrine cells in the mouse retina. Previous studies by other groups identified several roles for PTEN in retinal development, including neuron proliferation, migration and positioning, and neurite branching. However, these studies used conditional strategies that removed PTEN in retinal progenitors or newborn neurons, leaving open the question as to whether PTEN signaling regulates dendrite morphogenesis and synaptic connectivity in a cell intrinsic manner. To address this gap, Bracha and colleagues investigated PTEN regulation of dendrite morphogenesis and synaptic connectivity of the starburst amacrine cell dendrites, using the Chat-cre driver.

The authors report in detail the effects of PTEN conditional deletion on SAC morphology and functional contributions. The main findings are that PTEN-KO SACs have an excess number of small dendritic branches, while maintaining the overall morphological pattern. The authors defined the period of excess branching to occur between P14 and P21, which is past the normal period of dendritic branching and refinement, but this excess branching doesn't continue over maturity. PTEN-KO SACs also have enlarged somata and caliber of proximal dendrites, phenotypes commonly observed in PTEN-deficient neurons. Other structural features of SACs are unaffected, and there are no changes in the functional properties as measured by the direction selective responses of their target retinal ganglion cells.

Overall, the manuscript presents interesting PTEN loss-of-function effects on neuronal morphology. The manuscript is well written, the figures are high quality, and the data are rigorous and clearly reported. However, the morphological changes are very modest, and the interpretations and novel conceptual advances are not clear. For example, what is the significance of excess branching during the 2-3rd week, and does this inform on a novel developmental mechanism involving PTEN signaling. Second, some claims are not supported by data - for instance, measures of PTEN/PI3K signals are limited to pS6 and b-catenin Lef-GFP reporter but the study does not address whether these pathways contribute to the morphological phenotypes.

SUGGESTIONS TO AUTHORS

1. What is the significance of excess branching during the 2-3rd week, and does this inform on a novel developmental mechanism involving PTEN signaling? Do these findings suggest that PTEN signaling limits the developmental period of dendritic branching? Or maintains the mature structure? Or a consequence of hypertrophy?
2. Measures of PTEN/PI3K signals are limited to pS6 and b-catenin Lef-GFP reporter levels, but the study does not address whether these pathways contribute to the phenotypes. The authors suggest that 'increased dendrite branching is likely due to increased mTOR activity', but this is not tested. These signals remain high at P28, P60, when there is no excess branching. Pharmacological or genetic manipulations could address these claims.
3. The excess branching phenotype is interesting, but more details are needed to understand the impact on morphology. Does the excess branching cause an expansion in branch orders in the distal zone of the arbor - including a count of number of terminal branch endings in the distal zone of the arbor would clarify this question. The Scholl analysis shows no difference between GCL-SACs in WT and Pten cKO GCL-SAC at 90-100% radial length, which is perplexing given the increase in total branch points, and the SAC example in Figure 4. Or are the excess branches primarily small and terminal branches located within the proximal-50% zone (as in 3C"). The high mag images of INL-SAC in 3A"-3C" focus on small branches in the proximal zone. Is this main phenotype, and location of the excess branching?

4. mTorc2 contributes to cytoskeletal regulation and neuronal morphogenesis in some cell types. Testing the contributions of mTORC2 vs mTORC1 is likely beyond the scope of this study. However, their potential differential contributions should be raised in the discussion.
5. Chat-Cre can have a low recombination among the starburst population during postnatal development. Have the authors confirmed clearance of PTEN protein by immunostaining during the first postnatal week?

Minor point:

- Figure 1 should include a full view of the retinal layers of Pten; SixCre and Chat-Cre retinas, to contrast the phenotypes on the retinal layers and architecture.
- Figure 4 notes that ChATCre;Pten cKO SACs also had slightly smaller dendritic field areas compared to control. The authors should detail how they selected SAC locations to control for central-peripheral variation.

Reviewer 2

SUMMARY OF THE ADVANCE MADE IN THIS PAPER AND ITS POTENTIAL SIGNIFICANCE TO THE FIELD

Bracha et al. investigated the role of PTEN in starburst amacrine cells (SACs) by comparing pan-retinal and SAC-specific Pten deletion to dissect the PTEN's contribution to SAC development and circuit function. They found that SAC-specific deletion of Pten did not affect cell density, mosaic spacing, somatic positioning, or dendritic stratification-unlike the broad disorganization observed with pan-retinal Pten deletion. This suggested that PTEN is not required cell-autonomously for early SAC positioning or laminar targeting. Interestingly, the authors showed that loss of Pten in SACs leads to a progressive and persistent increase in dendritic branching by adulthood. These changes correlate with sustained activation of mTOR signaling, but not GSK3B, highlighting mTOR as a likely downstream effector. Notably, despite these morphological alterations, SACs maintain correct synaptic outputs, and postsynaptic direction-selective ganglion cells (DSGCs) exhibit normal direction selectivity. The measurements of SAC morphology are very detailed, and most conclusions are well supported by the data. Major and minor concerns are listed below.

SUGGESTIONS TO AUTHORS

1. The major weakness of this paper, perhaps, is the lack of functional changes in the Pten conditional knockout. Since both synaptic outputs and the direction-selectivity of DSGCs remain intact, the increased SAC dendritic branching appears to be dispensable for SAC synaptic and circuit functions. Therefore, the significance of Pten-mediated SAC dendritic branching is questionable.
2. The authors showed that elevated mTOR activity at P14-P60 in ChATCre; PtenCko knockout correlates with the increased dendritic branching, suggesting that mTOR activity may be a downstream mechanism. Testing whether inhibition of mTOR activity can rescue Pten-mediated SAC branching phenotype would strengthen the mechanistic conclusions.
3. In Figure 1B' and 1D', it is unclear whether the SACs indicated by the white arrows are Pten-positive.
4. In Figure 3, please include representative examples of self-crossing dendrites.

First revision

Author response to reviewers' comments

Dear Dr Silver,

Thank you for the opportunity to submit our revised study. We also thank the reviewers for their helpful comments and suggestions, and were pleased by their overall interest in the work. We have attempted to address each of their concerns in the revised manuscript, resulting in 4 new Supplemental Figures. In particular, both Reviewers pointed out that while our results showing increased pS6 levels from P14 to P60 suggest that mTOR signaling is involved in the branching phenotypes in *Pten*^{CKO} SACs, we did not demonstrate a causative role for mTORC in SAC morphological development. We have attempted to address this concern, and believe that our new data using pharmacological approaches provides additional support for mTORC's mechanistic role. We have also added additional text to the discussion to address some of the reviewer's points, as well as adding a "limitations" section to the end of the discussion. These text changes are indicated with red text. We hope that these revisions will address the reviewers' concerns in a satisfactory manner.

Our responses to the specific points raised by the reviewers follow below.

Reviewer 1:

1. What is the significance of excess branching during the 2-3rd week, and does this inform on a novel developmental mechanism involving PTEN signaling? Do these findings suggest that PTEN signaling limits the developmental period of dendritic branching? Or maintains the mature structure? Or a consequence of hypertrophy?

This is a good question, and we have added text to the discussion section to address the likely significance of this finding. In brief, we believe that PTEN signaling indeed limits that developmental period of dendritic branching, and that when we delete *Pten* from SACs, this window remains open for a longer period of time, resulting in excess branching. The fact that these branches persist at later ages (P60) and that that *Pten*-deficient SACs develop a hypertrophic branch suggests that PTEN also functions to maintain normal SAC morphology after development is complete.

2. Measures of PTEN/PI3K signals are limited to pS6 and b-catenin Lef-GFP reporter levels, but the study does not address whether these pathways contribute to the phenotypes. The authors suggest that 'increased dendrite branching is likely due to increased mTOR activity', but this is not tested. These signals remain high at P28, P60, when there is no excess branching. Pharmacological or genetic manipulations could address these claims.

We attempted to address this concern two different ways. First, we tried genetic manipulation. At an earlier stage of this study, we generated *ChAT*^{Cre};*Tsc2*^{CKO} mice in order to test the specific role of the mTORC pathway, but these mice were rather sick and did not survive past P14. During the revisions, we obtained *Tsc1*^{Flox} mice from Jackson Labs and generated *ChAT*^{Cre};*Tsc1*^{CKO} mice, but ran into the same issues: mice were born at normal Mendelian ratios, but mutants were small, failed to thrive, and died by P14 before we could conduct any meaningful analysis of SAC morphology. We do not know why the *Tsc1* and *Tsc2* conditional knockout mice have a more severe phenotype than *Pten* conditional knockouts, but we assume the severe phenotypes are due to defects in motor neurons that are targeted by the *ChAT*^{Cre} driver. We have included this information in the "Limitations" section of the discussion.

Since genetic manipulation did not work, we used a pharmacological approach: we injected mice with rapamycin (2.5 mg/kg, i.p.) every day for 2 weeks from P14 to P28, during the phase when SAC branching is most affected in *ChAT*^{Cre};*Pten*^{CKO} mice. We confirmed that this injection regimen led to decreased body weight in both control and cKO mice treated with Rapamycin compared to vehicle controls, and that it reduced the levels of pS6 in SACs in *ChAT*^{Cre};*Pten*^{CKO} mice, indicating that this dosing was sufficient to reduce mTOR activity in the mutants. Analysis of SAC morphology showed that Rapamycin dosing resulted in a partial rescue of SAC morphology at P28 in *ChAT*^{Cre};*Pten*^{CKO} mice. While this was not a complete rescue, possibly due to incomplete suppression of mTOR activity, it does support a role for the mTORC pathway SAC morphological development. This data is included in Supplemental Figure 6.

3. The excess branching phenotype is interesting, but more details are needed to understand the impact on morphology. Does the excess branching cause an expansion in branch orders in the distal

zone of the arbor - including a count of number of terminal branch endings in the distal zone of the arbor would clarify this question. The Scholl analysis shows no difference between GCL-SACs in WT and Pten cKO GCL-SAC at 90-100% radial length, which is perplexing given the increase in total branch points, and the SAC example in Figure 4. Or are the excess branches primarily small and terminal branches located within the proximal-50% zone (as in 3C"). The high mag images of INL-SAC in 3A"-3C" focus on small branches in the proximal zone. Is this main phenotype, and location of the excess branching?

As the Reviewer suggested, we conducted a branch level analysis, and this data is included as Supplemental Figure 2. We found that GCL SACs show an increase predominantly in lower level branches. This result is consistent with the excessive branching we see on the proximal dendrites close to the soma for both GCL and INL SACs. In INL SACs there is an overall increase branching across most of the branch level. It is not clear why there are subtle difference in branch level in between GCL and INL SACs; perhaps this is due to timing of *Chat^{Cre}* recombination or differences in the timing of normal GCL vs INL SAC branching dynamics.

4. mTorc2 contributes to cytoskeletal regulation and neuronal morphogenesis in some cell types. Testing the contributions of mTORC2 vs mTORC1 is likely beyond the scope of this study. However, their potential differential contributions should be raised in the discussion.

Thank you for this suggestion, and we agree that testing this directly would require additional mouse models that we currently do not have. We have now added a section discussing the potential differential contributions of mTORC1 and mTORC2 to the branching phenotypes in the discussion.

5. Chat-Cre can have a low recombination among the starburst population during postnatal development. Have the authors confirmed clearance of PTEN protein by immunostaining during the first postnatal week?

We have confirmed that PTEN protein is significantly reduced in SACs by P7 in the *Chat^{Cre};Pten^{CKO}* retinas using immunohistochemistry and include this data in Supplemental Figure 3.

Minor point:

- Figure 1 should include a full view of the retinal layers of Pten; SixCre and Chat-Cre retinas, to contrast the phenotypes on the retinal layers and architecture.

We have now included images of the full retina (Supplemental Figure 1) to contrast the layering and overall retinal architecture differences between the *Six3^{Cre};Pten^{CKO}* and *Chat^{Cre};Pten^{CKO}* retinas.

- Figure 4 notes that ChatCre;Pten cKO SACs also had slightly smaller dendritic field areas compared to control. The authors should detail how they selected SAC locations to control for central-peripheral variation

For our analysis, we quantified SACs from the central, medial, and peripheral areas of the retina, but excluded any SACs that were within 100mm of the optic nerve head or the edge of the retina. We did not record the precise location of individual SACs across all the experiments, but measurements include SACs from all areas. The materials and methods section has also been updated to include this information.

Reviewer 2:

1. The major weakness of this paper, perhaps, is the lack of functional changes in the Pten conditional knockout. Since both synaptic outputs and the direction-selectivity of DSGCs remain intact, the increased SAC dendritic branching appears to be dispensable for SAC synaptic and circuit functions. Therefore, the significance of Pten-mediated SAC dendritic branching is questionable.

We agree that the lack of functional changes in the *Chat^{Cre};Pten^{CKOs}* conditional knockouts is somewhat surprising, and agree that our results suggest that "increased SAC dendritic branching appears to be dispensable for SAC synaptic and circuit functions". However, we disagree that this

indicates that "*the significance of Pten-mediated SAC dendritic branching is questionable*". We think that our results are an important finding in light of the previous results showing major disruptions in overall retinal development in pan-retinal *Pten* conditional knockouts, and that they clarify the cell-autonomous role of PTEN-mediated signaling in SACs. We also did not directly record from SACs, so we cannot exclude that there may be more subtle functional defects in SAC or DS circuit function in the *Pten* conditional knockouts. We have highlighted this in the "Limitations" section of the discussion.

2. *The authors showed that elevated mTOR activity at P14-P60 in ChATCre; PtenCko knockout correlates with the increased dendritic branching, suggesting that mTOR activity may be a downstream mechanism. Testing whether inhibition of mTOR activity can rescue Pten-mediated SAC branching phenotype would strengthen the mechanistic conclusions.*

Please see the comments to Reviewer 1 above (Point #2)

3. *In Figure 1B' and 1D', it is unclear whether the SACs indicated by the white arrows are Pten-positive.*

We apologize for the lack of clarity in the original images. PTEN immunohistochemistry can be difficult to interpret because it appears to be present in every cell. We have now provided higher quality images in Figure 1, along with Supplemental Figure 2, and insets that highlight the presence of PTEN in control SACs (1B' and 1D') and loss of PTEN signal throughout the ganglion cell layer in *Six3^{Cre};Pten^{CKO}* retinas and specifically within SACs in the *ChAT^{Cre};Pten^{CKO}* retinas.

4. *In Figure 3, please include representative examples of self-crossing dendrites.*

We have now included insets to accompany the images in Figure 3 to highlight examples of dendrite self-crossing.

Second decision letter

MS ID#: dev.204980R1

MS TITLE: PTEN regulates starburst amacrine cell dendrite morphology during development

AUTHORS: Teva Bracha; Nina Luong; Joseph Leffler; Benjamin Sivyer; Kevin M. Wright

Dear Dr Wright,

I have now received all the referees reports on the above manuscript, and have reached a decision. The referees' comments are appended below.

The overall evaluation is positive and we would like to publish a revised manuscript in Development. Before we do so, please address the minor concern raised by reviewer 2 and detail them in your point-by-point response. If you do not agree with any of their criticisms or suggestions explain clearly why this is so. If it would be helpful, you are welcome to contact us to discuss your revision in greater detail. Please send us a point-by-point response indicating your plans for addressing the referees' comments, and we will look over this and provide further guidance.

Reviewer 1

SUMMARY OF THE ADVANCE MADE IN THIS PAPER AND ITS POTENTIAL SIGNIFICANCE TO THE FIELD

Focusing on the cholinergic starburst amacrine cells in the mouse retina. this study demonstrates that loss of PTEN leads to increased branching, branch and soma hypertrophy, and overall maintenance of dendritic arborizations in this neuron model.

The authors have addressed all of my points, added new data, and attempted to investigate other mouse lines with altered mTor signaling, but these lines had early endpoints. The authors show that rapamycin treatment has a trending reduction on decreasing branch number, supporting a contribution of elevated mTor signaling in this phenotype, but the effect is not significant. This is likely related to the main limitation of this study where PTEN loss in maturing starburst neurons produces mild morphological phenotypes, and has no effect on starburst-dependent functional outputs that shape direction selectivity measured by MEAs. Nevertheless, the main conceptual advance is significant and this is what retains my enthusiasm: elevated signaling resulting from Pten deletion in the 2-3 postnatal weeks promotes ongoing branching when the normal developmental period has likely ceased, which bears important insights into consequences of Pten deficiencies and will inform the field. The authors have improved the contextualization of the results and potential significance in the discussion.

SUGGESTIONS TO AUTHORS

I only have a minor points requiring clarification in the text:

Figure title/Results for Figure S6 are described as 'partially rescues' or 'partially attenuate', which is ambiguous and reflects an interpretation. The data show a trend in reduced branching but are not significant. The authors should state these results clearly in the Results text, then offer interpretation such as 'attenuate'. The figure legend title should be changed.

The reporting for the p values for the pair-wise comparisons in Figure S6 is confusing. It would be helpful to see the p values for the pair of interest on the graph (M, control cko vs rapa + cko), or use the same terminology shown on the graph in the legend.

Second revision

Author response to reviewers' comments

Dear Dr Silver,

Thank you for the provisional acceptance of our manuscript. We have addressed the minor comment from Reviewer 1 in the revised version. We have changed the title of Supplemental Figure 6, and have made appropriate corresponding text changes related to the data in this figure to more accurately reflect the results. We hope that these revisions will address the reviewers' concerns in a satisfactory manner.

Sincerely,

Kevin Wright

Third decision letter

MS ID#: dev.204980R2

MS TITLE: PTEN regulates starburst amacrine cell dendrite morphology during development

AUTHORS: Teva Bracha; Nina Luong; Joseph Leffler; Benjamin Sivyer; Kevin M. Wright

Dear Dr Wright,

I am happy to tell you that your manuscript has been accepted for publication in Development, pending our standard publication integrity checks.